# TAK1-mediated phosphorylation of PLCE1 represses PIP2 hydrolysis to impede esophageal squamous cancer metastasis

Qianqian Ju[1], Wenjing Sheng[1], Meichen Zhang[1], Jing Chen[1], Liucheng Wu[2], Xiaoyu Liu[1], Wentao Fang[3]*, Hui Shi[3]*, Cheng Sun[1]*

[1]Key Laboratory of Neuroregeneration of Jiangsu and Ministry of Education; NMPA Key Laboratory for Research and Evaluation of Tissue Engineering Technology Products; School of Medicine, Nantong University, Nantong, China; [2]Laboratory Animal Center, Nantong University, Nantong, China; [3]Department of Thoracic Surgery, Shanghai Chest Hospital, Shanghai Jiao Tong University School of Medicine, Shanghai, China

**\*For correspondence:**
vwtfang@hotmail.com (WF);
shihui1@shchest.org (HS);
suncheng1975@ntu.edu.cn (CS)

**Competing interest:** The authors declare that no competing interests exist.

## eLife assessment

This work provides **solid** evidence that Transforming Growth Factor β Activated Kinase 1 (TAK1) regulates esophageal squamous cell carcinoma (ESCC) tumor proliferation and metastasis. The findings are **valuable** to the field of molecular tumor biology in general and to the understanding of ESCC tumor invasiveness and metastatic potential.

**Abstract** TAK1 is a serine/threonine protein kinase that is a key regulator in a wide variety of cellular processes. However, the functions and mechanisms involved in cancer metastasis are still not well understood. Here, we found that TAK1 knockdown promoted esophageal squamous cancer carcinoma (ESCC) migration and invasion, whereas TAK1 overexpression resulted in the opposite outcome. These in vitro findings were recapitulated in vivo in a xenograft metastatic mouse model. Mechanistically, co-immunoprecipitation and mass spectrometry demonstrated that TAK1 interacted with phospholipase C epsilon 1 (PLCE1) and phosphorylated PLCE1 at serine 1060 (S1060). Functional studies revealed that phosphorylation at S1060 in PLCE1 resulted in decreased enzyme activity, leading to the repression of phosphatidylinositol 4,5-bisphosphate (PIP2) hydrolysis. As a result, the degradation products of PIP2 including diacylglycerol (DAG) and inositol IP3 were reduced, which thereby suppressed signal transduction in the axis of PKC/GSK-3β/β-Catenin. Consequently, expression of cancer metastasis-related genes was impeded by TAK1. Overall, our data indicate that TAK1 plays a negative role in ESCC metastasis, which depends on the TAK1-induced phosphorylation of PLCE1 at S1060.

## Introduction

Esophageal cancer (EC) is the seventh most common malignancy worldwide, with more than 600,000 new cases diagnosed annually (*Sung et al., 2021*). The primary histological subtypes of EC include esophageal squamous cell carcinoma (ESCC) and esophageal adenocarcinoma (*He et al., 2021; Lagergren et al., 2017*). ESCC is a major subtype of EC in China, accounting for more than 90%

of EC (*He et al., 2021*). The early diagnosis rate of ESCC is very low owing to the lack of specific biomarkers. Thus, most patients are found to be in locally advanced or metastatic stages when they are diagnosed. The 5-year survival rate is approximately 15–25% (*Pennathur et al., 2013*). In recent decades, although therapeutic strategies have made great progress toward ESCC, recurrence and metastasis remain significant challenges (*Mao et al., 2021*). Therefore, there is an urgent need to deeply understand the mechanisms involved in cell metastasis, and thus try to explore new druggable targets for treating ESCC.

Transforming growth factor β-activated kinase 1 (TAK1), a member of the mitogen-activated protein kinase (MAPK) kinase kinase (MAP3K) family, is encoded by the gene *Map3k7*. As a serine/threonine protein kinase, TAK1 has been shown to play an integral role in inflammatory signal transduction through multiple pathways (*Sakurai, 2012*). TAK1 is involved in a wide variety of cellular processes, including cell survival, cell migration and invasion, inflammation, immune regulation, and tumorigenesis (*Cho et al., 2021*; *Mukhopadhyay and Lee, 2020*). Although the role of TAK1 has been widely studied, its precise role remains controversial. For example, by acting as a tumor suppressor, TAK1 has been shown to negatively associated with tumor progression in several human cancers including prostate cancer (*Huang et al., 2021*), hepatocellular carcinoma (HCC) (*Tan et al., 2020*; *Wang et al., 2021*), cervical cancer (*Guan et al., 2017*), and certain blood cancers (*Guo et al., 2019*). In contrast, TAK1 promotes tumor progression in a variety of human cancers, including colon, ovarian, lung, and breast cancers (*Augeri et al., 2016*; *Cai et al., 2014*; *Xu et al., 2022*). These aforementioned findings indicate that TAK1 plays a dual role in tumor initiation, progression, and metastasis. In our previous study, TAK1 has been shown to phosphorylate RASSF9 at serine 284 to inhibit cell proliferation by targeting the RAS/MEK/ERK axis in ESCC (*Shi et al., 2021*). To date, there have been no reports on the precise role of TAK1 in ESCC metastasis.

Phospholipase C epsilon 1 (PLCE1) is encoded by the *Plce1* gene on chromosome 10q23 in human and belongs to the phospholipase C (PLC) family (*Fukami et al., 2010*; *Kadamur and Ross, 2013*). Like other PLC family members, PLCE1 is composed of a PLC catalytic domain, PH domain, EF domain, and C2 domain. In addition, PLCE1 contains unique regions, two C-terminal Ras association (RA) domains and an N-terminal CDC25-homology domain (*Kadamur and Ross, 2013*). Once activated, PLCE1 is essential for intracellular signaling by catalyzing the hydrolysis of membrane phospholipids such as phosphatidylinositol 4,5-bisphosphate (PIP2), in order to produce two important secondary messengers, inositol 1,4,5-trisphosphate (IP3) and diacylglycerol (DAG), which further trigger IP3-dependent calcium ion ($Ca^{2+}$) release from the endoplasmic reticulum and PKC activation (*Fukami et al., 2010*; *Kadamur and Ross, 2013*). Accumulating evidence has shown that PLCE1 promotes cell growth, migration, and metastasis in multiple human cancers, including hepatocellular carcinoma, non-small-cell lung cancer, head and neck cancer, bladder cancer, gastric cancer, and prostate cancer (*Abnet et al., 2010*; *Fan et al., 2019*; *Liao et al., 2017*; *Ma et al., 2011*; *Ou et al., 2010*; *Wang et al., 2020*; *Yue et al., 2019*). However, whether and how PLCE1 affects cancer metastasis in ESCC remain largely unknown.

In the present study, we have examined the potential role of TAK1 in ESCC metastasis. We found that TAK1 negatively regulates cell migration and invasion in ESCC, and that PLCE1 is a downstream target of TAK1. TAK1 phosphorylates PLCE1 at serine 1060 (S1060) to inhibit its enzymatic activity, leading to decreased IP3 and DAG levels, both of which are products of PLCE1 catalyzed reactions. As a result, IP3/DAG triggered signal transduction in the axis of PKC/GSK-3β/β-Catenin was blunted. All these effects induced by TAK1 resulted in the repression of cell migration and invasion in ESCC.

## Results

### TAK1 negatively regulates ESCC migration and invasion

In our previous study, we showed that TAK1 expression was reduced in esophageal squamous tumor tissues, and that TAK1 inhibits ESCC proliferation (*Shi et al., 2021*). To examine whether TAK1 affects the epithelial-mesenchymal transition (EMT) process in ESCC, we increased TAK1 expression in ECA-109 cells by transfecting a plasmid expressing *Map3k7* (TAK1 gene name) and confirmed the overexpression of TAK1 (*Figure 1A*). Owing to the promising role of epidermal growth factor (EGF) in EMT (*Lu et al., 2003*), it has been used to trigger EMT in ESCC. As shown in *Figure 1B*, EGF treatment induced a spindle-shaped morphology in ECA-109 cells, which was markedly prevented by TAK1.

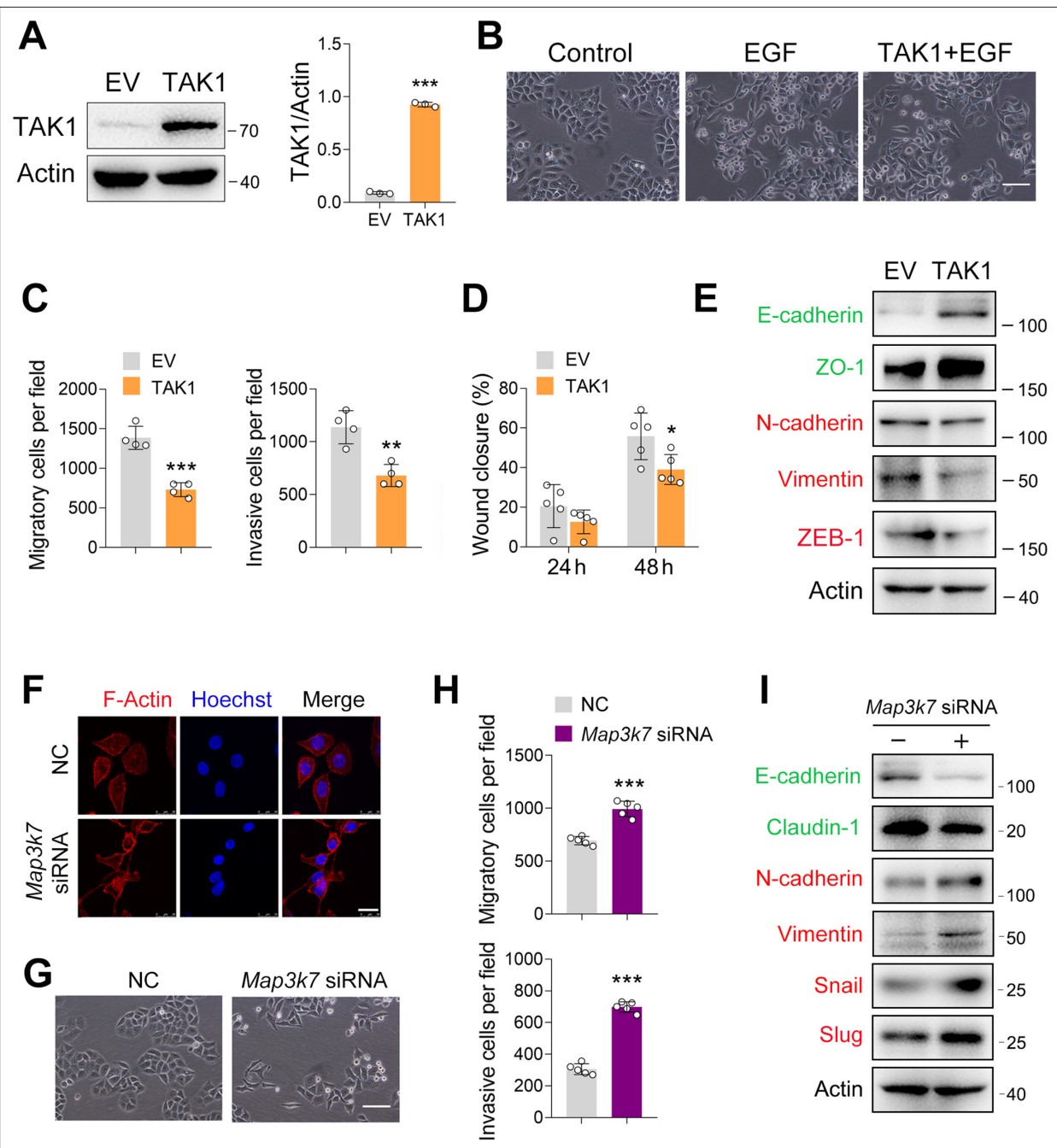

**Figure 1.** TAK1 negatively regulates esophageal squamous cell carcinoma (ESCC) migration and invasion. (**A**) Increased expression of TAK1 in ECA-109 cells transfected with a plasmid expressing *Map3k7*. (**B**) Increased expression of TAK1 inhibits the morphological changes to form spindle-shaped mesenchymal cells induced by epidermal growth factor (EGF) (100 ng/ml) in ECA-109. Scale bar = 100 μm. (**C**) Increased expression of TAK1 inhibits cell migration and invasion in ECA-109 cells. Cell migration and invasion were analyzed by transwell assay. n=4 biologically independent replicates. (**D**) Wound healing assay showing cell migration was attenuated by TAK1. n=5 biologically independent replicates. (**E**) TAK1 decreased mesenchymal marker gene expression, while increased the expression of epithelial markers. ECA-109 cells were transfected with the plasmid carrying Map3k7. 24 hr post-transfection, protein samples were prepared and subjected to western blot. Actin was used as a loading control. (**F**) Knockdown of TAK1 increased the expression of F-Actin. ECA-109 cells were transfected with Map3k7 siRNA. 72 hr post-transfection, cells were subjected to immunofluorescence analysis using an anti-F-Actin antibody (red). Hoechst was used to stain the nucleus (blue). Scale bar = 10 μm. (**G**) TAK1 knockdown induces spindle-shaped mesenchymal cell morphology in ECA-109 cells. Scale bar = 100 μm. (**H**) Reduced expression of TAK1 promotes cell migration and invasion. ECA-109 cells were transfected with Map3k7 siRNA. 72 hr post-transfection, cell migration and invasion were analyzed by transwell assay. n=5 biologically independent replicates. (**I**) Knockdown of TAK1 increases mesenchymal protein marker expression and decreases epithelial protein marker expression. Data are presented as mean ± SD. Statistical significance was tested by unpaired Student's t-test. *p<0.05, **p<0.01, and ***p<0.001.

*Figure 1 continued on next page*

*Figure 1 continued*

The online version of this article includes the following source data and figure supplement(s) for figure 1:

**Source data 1.** TAK1 negatively regulates esophageal squamous cell carcinoma (ESCC) migration and invasion.

**Source data 2.** PDF file containing original western blots for *Figure 1A, E, and I*, indicating the relevant bands.

**Source data 3.** Original files for western blot analysis displayed in *Figure 1A, E, and I*.

**Figure supplement 1.** TAK1 represses cell migration in ECA-109 cells.

**Figure supplement 1—source data 1.** TAK1 represses cell migration in ECA-109 cells.

**Figure supplement 2.** TAK1 silencing promotes esophageal squamous cell carcinoma (ESCC) migration and invasion.

**Figure supplement 2—source data 1.** TAK1 silencing promotes esophageal squamous cell carcinoma (ESCC) migration and invasion.

**Figure supplement 3.** TAK1 knockdown facilitates esophageal squamous cell carcinoma (ESCC) migration and invasion.

**Figure supplement 3—source data 1.** TAK1 knockdown facilitates esophageal squamous cell carcinoma (ESCC) migration and invasion.

**Figure supplement 3—source data 2.** PDF file containing original western blots for *Figure 1—figure supplement 3B and F*, indicating the relevant bands.

**Figure supplement 3—source data 3.** Original files for western blot analysis displayed in *Figure 1—figure supplement 3B and F*.

**Figure supplement 4.** TAK1 knockout accelerates esophageal squamous cell carcinoma (ESCC) migration and invasion.

**Figure supplement 4—source data 1.** TAK1 knockout accelerates esophageal squamous cell carcinoma (ESCC) migration and invasion.

**Figure supplement 4—source data 2.** PDF file containing original western blots for *Figure 1—figure supplement 4A and E*, indicating the relevant bands.

**Figure supplement 4—source data 3.** Original files for western blot analysis displayed in *Figure 1—figure supplement 4A and E*.

**Figure supplement 5.** Inhibition of TAK1 potentiates cell migration and invasion in ECA-109 cells.

**Figure supplement 5—source data 1.** Inhibition of TAK1 potentiates cell migration and invasion in ECA-109 cells.

These data implied that TAK1 is a negative regulator of EMT in ESCC. To address this hypothesis, we performed a transwell assay and found that TAK1 repressed the migration and invasion of ECA-109 cells (*Figure 1C*; *Figure 1—figure supplement 1A*). The wound healing assay further confirmed the negative effects of TAK1 on cancer cell migration (*Figure 1D*; *Figure 1—figure supplement 1B*). Next, we examined whether TAK1 affects EMT-related gene expression. Our data showed that TAK1 increased E-cadherin and ZO-1, two epithelial molecules, whereas mesenchymal molecules such as N-cadherin, Vimentin, and ZEB1 were reduced by TAK1 (*Figure 1E*; *Figure 1—figure supplement 1C*). The quantitative real time-PCR (qRT-PCR) data confirmed these changes (*Figure 1—figure supplement 1D*).

Next, we performed loss-of-function experiments to examine the effect of TAK1 on ESCC cell migration and invasion. The knockdown efficiency of *Map3k7* siRNA on TAK1 expression has been reported previously (*Shi et al., 2021*). As EMT involves dynamic and spatial regulation of the cytoskeleton (*Fife et al., 2014*; *Li and Wang, 2020*), we analyzed the expression of F-Actin by immunofluorescence. As shown in *Figure 1F*, TAK1 knockdown induced F-Actin expression. TAK1 knockdown promoted a spindle-shaped mesenchymal morphology in ECA-109 cells (*Figure 1G*). In accordance with these changes, cell invasion and migration were stimulated, as evidenced by data from the wound healing and transwell assays (*Figure 1H*; *Figure 1—figure supplement 2A and B*). The epithelial markers including E-cadherin and Claudin-1 were found to have decreased by TAK1 knockdown, whereas mesenchymal markers N-cadherin, Vimentin, Snail, and Slug were increased (*Figure 1I*; *Figure 1—figure supplement 2C*). At the mRNA level, *Cdh1* and *Cldn1* were downregulated by TAK1 knockdown, whereas *Cdh2* and *Snail1* were upregulated (*Figure 1—figure supplement 2D*). To verify the effect of TAK1 knockdown on cell migration and invasion, we downregulated TAK1 expression using a lentivirus carrying *Map3k7* shRNA (LV-*Map3k7* shRNA) (*Figure 1—figure supplement 3A and B*). In LV-*Map3k7* shRNA-transduced cells, cell migration and invasion were clearly increased (*Figure 1—figure supplement 3C–E*). The epithelial markers E-cadherin, ZO-1, and Claudin-1 were repressed by TAK1 knockdown, whereas mesenchymal markers N-cadherin, Slug, and ZEB-1 were activated (*Figure 1—figure supplement 3F and G*). Our qRT-PCR data showed similar changes in these EMT genes (*Figure 1—figure supplement 3H*). Additionally, TAK1 knockdown was accomplished by CRISPR-Cas9 using *Map3k7* guide RNA (gRNA) (*Figure 1—figure supplement 4A*). Again, we observed that cell migration and invasion were enhanced by *Map3k7* gRNA (*Figure 1—figure supplement*

*4B-D*). The expression of E-cadherin, ZO-1, and Claudin-1 was inhibited by *Map3k7* gRNA, whereas the expression of N-cadherin, Vimentin, Snail, and Slug was activated (*Figure 1—figure supplement 4E and F*). Similar changes in these EMT-related gene expression were observed using qRT-PCR (*Figure 1—figure supplement 4G*). Furthermore, we inhibited TAK1 activity using Oxo, NG25, or Takinib and observed that all of these treatments induced morphological changes in ECA-109 cells from round shapes to spindle-like shapes (*Figure 1—figure supplement 5A*). Cell migration and invasion were also stimulated by TAK1 inhibition (*Figure 1—figure supplement 5B–E*). Taken together, these findings indicate that TAK1 negatively regulates ESCC migration and invasion.

## TAK1 phosphorylates PLCE1 at serine 1060

These aforementioned data clearly show that TAK1 negatively regulates the cell migration and invasion of ESCC. To reveal the underlying mechanism, we performed co-immunoprecipitation combined with mass spectrometry in order to identify potential downstream targets of TAK1. As previously reported, 24 proteins were phosphorylated in the immunocomplex (*Shi et al., 2021*). Of these, phospholipase C epsilon 1 (PLCE1) has attracted our attention because of its essential role in cell growth, migration, and metastasis in various human cancers (*Abnet et al., 2010*; *Chen et al., 2019*; *Chen et al., 2020*; *Gu et al., 2018*; *Kadamur and Ross, 2013*; *Wang et al., 2010*). According to the mass spectrometry data, the serine residue at position 1060 (S1060) in PLCE1 was phosphorylated (*Figure 2A*). Currently, there is no antibody against phosphorylated PLCE1 at S1060 (p-PLCE1 S1060), we therefore generated an antibody to detect p-PLCE1 S1060. To verify antibody specificity, S1060 in PLCE1 was mutated to alanine (S1060A). As shown in *Figure 2B*, wildtype (WT) PLCE1 was phosphorylated by TAK1 at S1060, whereas PCLE1 S1060A was resistant to TAK1-induced phosphorylation at S1060. Moreover, the p-PLCE1 S1060 induced by TAK1 was weakened by(5Z)-7-oxozeaenol (Oxo), a potent inhibitor of TAK1 (*Figure 2C*). Notably, phosphorylated TAK1 (p-TAK1) markedly decreased in the presence of Oxo, indicating that TAK1 was inhibited (*Figure 2C*). To further confirm these findings, other TAK1 inhibitors, such as Takinib and NG25, were used, and similar changes in p-PLCE1 and p-TAK1 were observed (*Figure 2D and E*). As for endogenous p-PLCE1 S1060, it was increased and decreased, respectively, by TAK1 overexpression and knockdown (*Figure 2—figure supplement 1A and B*). Moreover, the in vitro kinase assay showed that PLCE1 was phosphorylated by TAK1 at serine 1060 (*Figure 2—figure supplement 1C*). Collectively, these results indicate that PLCE1 is a downstream target of TAK1, and that PLCE1 S1060 is phosphorylated by TAK1. To further confirm this notion, we analyzed TAK1 and p-PCLE1 expression in clinical samples. As shown in *Figure 2F*, TAK1 expression was reduced in tumor tissues compared to that in their respective adjacent normal tissues. In accordance with these changes in TAK1, p-PLCE1 levels were also decreased in tumor tissues (*Figure 2G*) Pearson's correlation tests showed that TAK1 expression was positively correlated with p-PLCE1 expression (*Figure 2H*). These changes in TAK1 and p-PLCE1 levels were confirmed by western blot data (*Figure 2I*).

## TAK1 phosphorylates PLCE1 to inhibit cell migration and invasion in ESCC

It has been well documented that PLCE1 plays a key role in cancer progression (*Abnet et al., 2010*; *Chen et al., 2019*; *Chen et al., 2020*; *Gu et al., 2018*; *Wang et al., 2010*). Therefore, we predicted that the negative effects of TAK1 on ESCC migration and invasion may rely on PLCE1. To verify this hypothesis, we first examined whether PLCE1 affects cell migration and invasion. In ECA-109 cells, PLCE1 overexpression significantly increased the migration and invasion (*Figure 3A–C*; *Figure 3—figure supplement 1A and B*). In accordance with these changes, the epithelial marker E-cadherin was repressed by PLCE1, whereas the mesenchymal markers including Vimentin, Snail, Slug, and ZEB-1 were activated (*Figure 3D*; *Figure 3—figure supplement 1C*). The qRT-PCR data also revealed that Cdh1 was downregulated by PLCE1, while Vim, Snail1, and Snail2 were upregulated (*Figure 3—figure supplement 1D*). To confirm these findings, we knocked down PLCE1 using siRNA. As shown in *Figure 3E*, Plce1 expression was markedly reduced by all three tested siRNAs. Similar changes were observed in the western blot data (*Figure 3F*). Among these siRNAs, siRNA-2# was found to exhibit the highest knockdown efficiency and was selected for subsequent experiments. On the contrary, as compared to PLCE1 overexpression, PLCE1 knockdown impeded cell migration and invasion (*Figure 3G and H*; *Figure 3—figure supplement 2A and B*). The expression patterns of genes

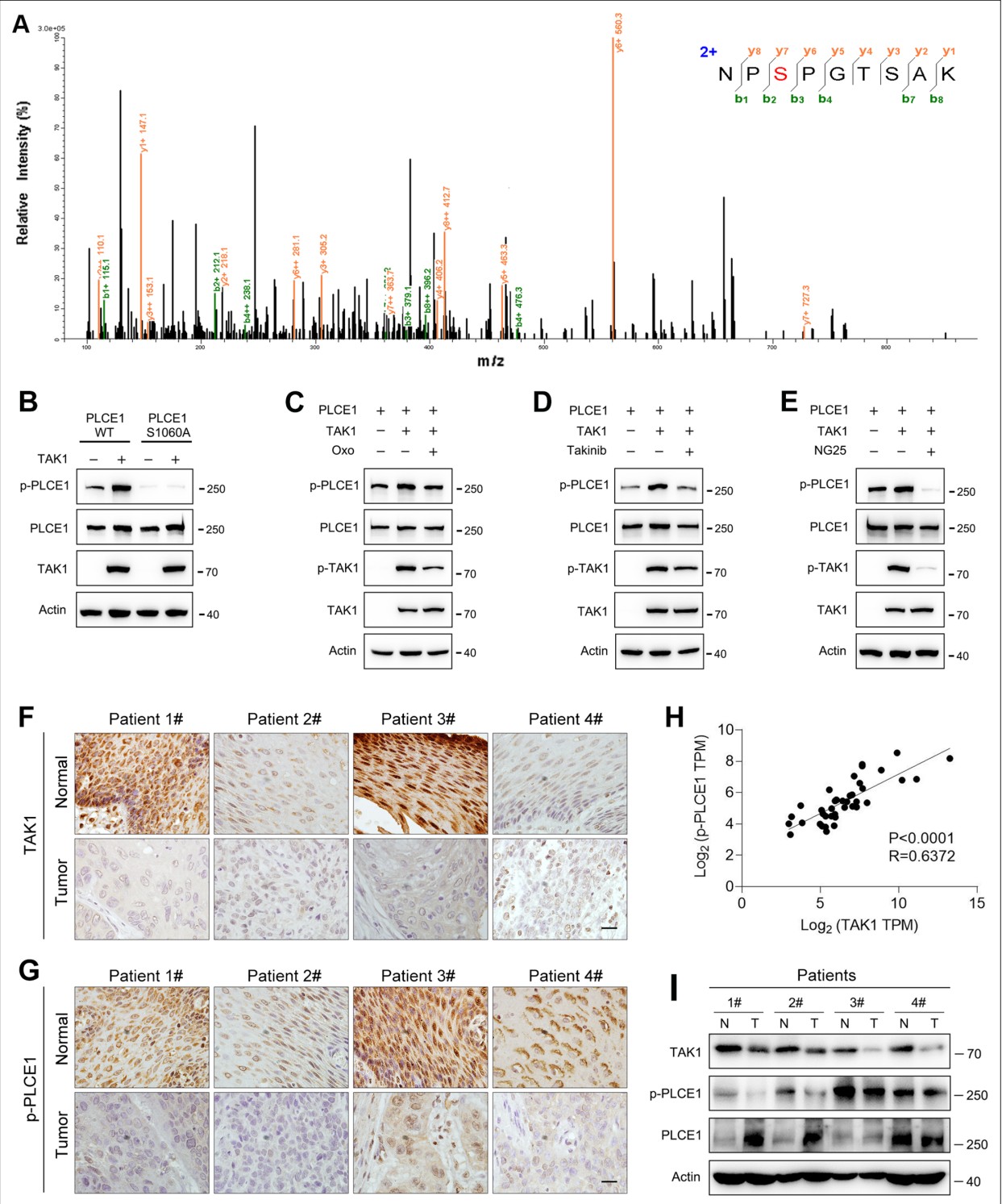

**Figure 2.** TAK1 phosphorylates PLCE1 at serine 1060. (**A**) Tandem mass spectrometry showing serine 1060 (S1060) in PLCE1 was phosphorylated by TAK1. ECA-109 cells were transfected with a plasmid expressing *Map3k7*. 24 hr post-transfection, cells were harvested and subjected to co-immunoprecipitation. The resulting immunocomplex was analyzed by liquid chromatography coupled with tandem mass spectrometry (LC-MS/MS). (**B**) TAK1 fails to phosphorylate PLCE1$^{S1060A}$. ECA-109 cells were co-transfected with the plasmids carrying wildtype (WT) *Plce1*, mutated *Plce1* (PLCE1$^{S1060A}$), or *Map3k7* as indicated. 24 hr post-transfection, cells were collected for western blot analysis. (**C–E**) Inhibition of TAK1 reduces PLCE1 phosphorylation at S1060. ECA-109 cells were co-transfected with the plasmids expressing *Plce1* or *Map3k7*. 6 hr post-transfection, TAK1 inhibitor (5Z)-7-oxozeaenol (Oxo; 10 μM) (**C**), or 10 μM Takinib (**D**), or 10 μM NG25 (**E**) was added in culture medium, and cells were cultured for additional 18 hr. Cells were then subjected to western blot analysis. Actin was used as a loading control. (**F–G**) Immunohistochemical analysis of TAK1 (**F**) and p-PLCE1

*Figure 2 continued on next page*

*Figure 2 continued*

(**G**) expression in normal and esophageal squamous tumor tissues. n=4 biologically independent replicates. Scale bar = 20 μm. (**H**) Correlation between p-PLCE1 and TAK1 based on immunohistochemical data as shown in (**F–G**). 10 views for each sample were randomly chosen for Pearson's correlation test. (**I**) TAK1 and p-PLCE1 protein levels in clinical samples. Protein levels were analyzed by western blot, and Actin was used as a loading control. n=4 biologically independent replicates. N: normal tissue; T: tumor tissue.

The online version of this article includes the following source data and figure supplement(s) for figure 2:

**Source data 1.** TAK1 phosphorylates PLCE1 at serine 1060.

**Source data 2.** PDF file containing original western blots for *Figure 2B, C, D, E, and I*, indicating the relevant bands.

**Source data 3.** Original files for western blot analysis displayed in *Figure 2B, C, D, E, and I*.

**Figure supplement 1.** TAK1 phosphorylates PLCE1 at serine 1060.

**Figure supplement 1—source data 1.** PDF file containing original western blots for *Figure 2—figure supplement 1A, B, and C*, indicating the relevant bands.

**Figure supplement 1—source data 2.** Original files for western blot analysis displayed in *Figure 2—figure supplement 1A, B, and C*.

involved in EMT were contrary to those of PLCE1 overexpression (*Figure 3I*; *Figure 3—figure supplement 2C*). Moreover, similar to PLCE1 overexpression, the activation of PLCE1 by m-3M3FBS potentiated cell migration and invasion (*Figure 3—figure supplement 3*). PLCE1 inhibition by U-73122 recapitulated the phenotypes induced by PLCE1 knockdown (*Figure 3—figure supplement 4*).

Since PLCE1 is a lipid hydrolase, we next investigated whether TAK1-induced phosphorylation of PLCE1 affects its enzymatic activity. To this end, we transfected ECA-109 cells with a plasmid bearing Myc-PLCE1 or TAK1, and PLCE1 was captured by pull-down using anti-Myc beads, which was then subjected to enzyme activity assay. As shown in *Figure 4A*, the PLCE1 activity was reduced in the presence of TAK1; however, the PLCE1 S1060A activity was not affected (*Figure 4B*), indicating TAK1-induced phosphorylation at S1060 repressed PLCE1 activity. As a lipid hydrolase, PLCE1 catalyzes PIP2 hydrolysis to produce IP3 and DAG, both of which are secondary messengers involved in diverse cellular processes (*Kadamur and Ross, 2013*). Therefore, we examined these two products to verify the inhibitory effect of TAK1 on PLCE1 expression. Our data showed that the productions of IP3 and DAG were increased by PLCE1, which was largely counteracted by TAK1 (*Figure 4C and E*). As a messenger, IP3 induces the endoplasmic reticulum to release $Ca^{2+}$ into the cytoplasm (*Kadamur and Ross, 2013*). Hence, we analyzed cytoplasmic $Ca^{2+}$ levels using Fluo-4 AM staining. As expected, the signals for the cytoplasmic $Ca^{2+}$ were increased by PLCE1 in ECA-109 cells, and this trend was attenuated by TAK1 (*Figure 4F and G*). We also directly detected cytoplasmic $Ca^{2+}$ using a fluorospectrophotometer and observed similar changes (*Figure 4H*). Flow cytometry analysis also evidenced the increase in the cytoplasmic $Ca^{2+}$ induced by PLCE1 was prevented by TAK1 (*Figure 4I*). This evidence indicates that TAK1 inhibits PLCE1 activity, which is likely due to TAK1-induced phosphorylation of PLCE1 at S1060. In order to further verify this hypothesis, a series of functional experiments was performed. For instance, we observed that TAK1 reversed PLCE1-induced cell migration and invasion; however, TAK1 failed to affect PLCE1 S1060A-induced cell migration and invasion (*Figure 4—figure supplement 1*). In addition to PLCE1, some other signaling pathways were also activated by TAK1. For example, TAK1 overexpression induced increases in p-IKK, p-JNK, p-p38 MAPK, and p-ERK in ECA-109 cells, whereas TAK1 knockdown reduced these protein levels (*Figure 4—figure supplement 2*). At this regard, PLCE1 is likely a unique substrate of TAK1 for transducing its inhibitory effects on ESCC migration and invasion, although some other signaling pathways are also affected by TAK1. Overall, these data clearly indicate that PLCE1 positively regulates ESCC migration and invasion. By inducing phosphorylation at S1060, TAK1 inhibits PLCE1 enzyme activity, thereby counteracting PLCE1-induced ESCC migration and invasion. PLCE1 is a downstream target of TAK1 that inhibits ESCC migration and invasion.

## TAK1 inhibits PLCE1-induced signal transduction in the PKC/GSK-3β/β-Catenin axis

PLCE1 hydrolyzes PIP2 to produce two important secondary messengers, DAG and IP3, which trigger $Ca^{2+}$ release from the endoplasmic reticulum into the cytoplasm to activate PKC (*Harden et al., 2009*; *Kadamur and Ross, 2013*), suggesting that PKC activation is responsible for PLCE1-induced cell migration and invasion in ESCC. In order to test this hypothesis, we treated cells with 2-APB, an IP3

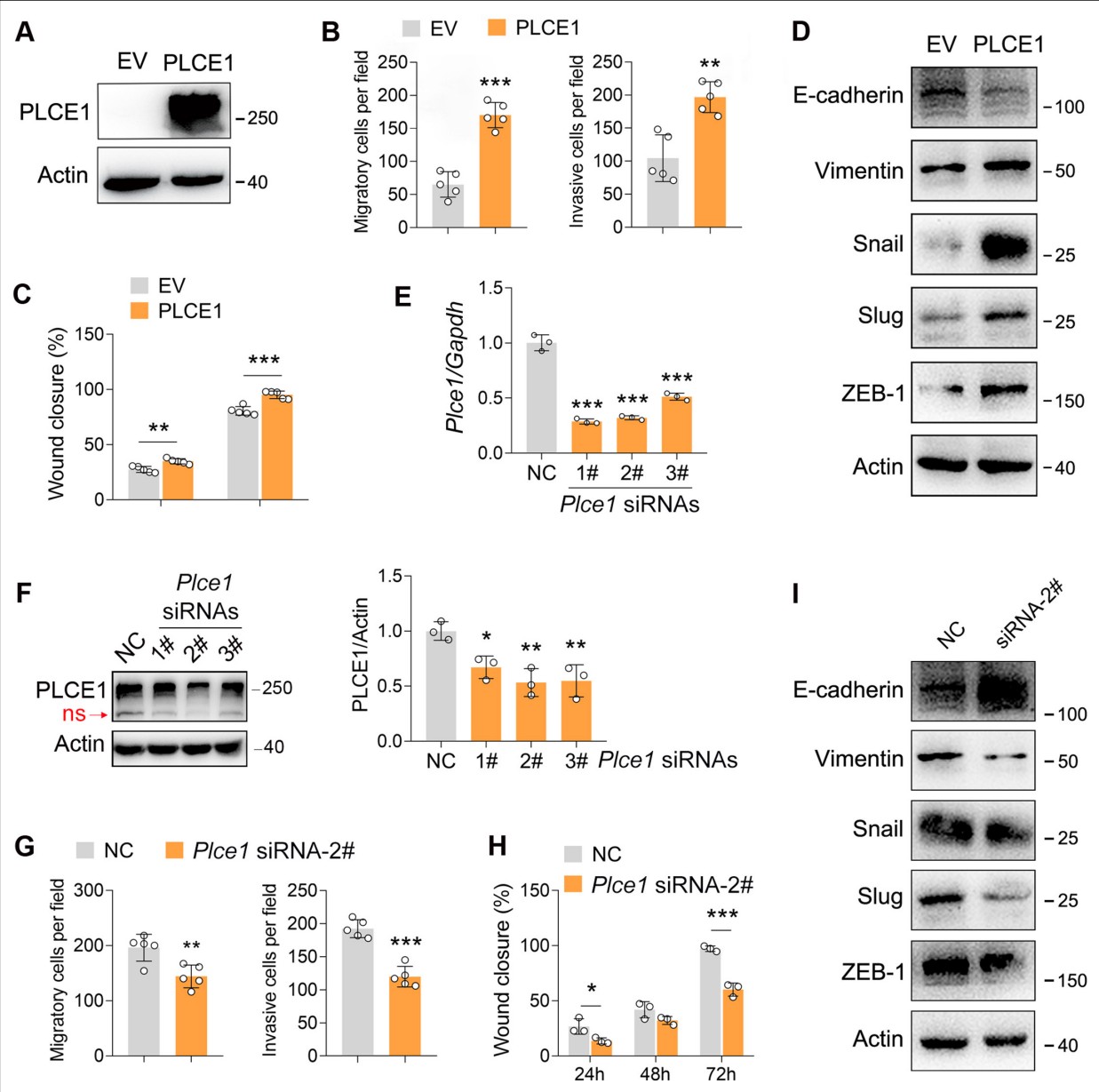

**Figure 3.** PLCE1 positively regulates esophageal squamous cell carcinoma (ESCC) migration and invasion. (**A**) Increased expression of PLCE1 in ECA-109 cells transfected with a plasmid expressing *Plce1*. (**B–C**) Increased expression of PLCE1 enhances cell migration and invasion. ECA-109 cells were transfected with the plasmid carrying *Plce1*. 24 hr post-transfection, cells were subjected to transwell (**B**) or wound healing (**C**) assay. n=5 biologically independent replicates. (**D**) Increased expression of PLCE1 in ECA-109 cells induces mesenchymal protein marker expression, while reduces epithelial protein marker expression. (**E–F**) Knockdown of PLCE1. ECA-109 cells were transfected with siRNAs targeting *Plce1*. 72 hr post-transfection, cells were harvested for analyzing PLCE1 expression by quantitative real time-PCR (qRT-PCR) (**E**) and western blot (**F**). (**G–H**) Reduced expression of PLCE1 inhibits cell migration and invasion. ECA-109 cells were transfected with *Plce1* siRNA-2. 48 hr post-transfection, cells were subjected to transwell (**G**) or wound healing (**H**) assay. n=3–5 biologically independent replicates. (**I**) Knockdown of PLCE1 promotes epithelial protein marker expression, while represses mesenchymal protein marker expression. ECA-109 cells were transfected with *Plce1* siRNA-2. 72 hr post-transfection, cells were harvested and subjected to western blot analysis. Protein level was detected by western blot, and Actin was used as a loading control. Gene expression was analyzed by qRT-PCR, and *Gapdh* was used as a house-keeping gene. Data are presented as mean ± SD. Statistical significance was tested by unpaired Student's t-test. *p<0.05, **p<0.01, and ***p<0.001.

The online version of this article includes the following source data and figure supplement(s) for figure 3:

**Source data 1.** PLCE1 positively regulates esophageal squamous cell carcinoma (ESCC) migration and invasion.

**Source data 2.** PDF file containing original western blots for *Figure 3A, D, F, and I*, indicating the relevant bands.

**Source data 3.** Original files for western blot analysis displayed in *Figure 3A, D, F, and I*.

*Figure 3 continued on next page*

*Figure 3 continued*

**Figure supplement 1.** PLCE1 promotes cell migration and invasion in ECA-109 cells.

**Figure supplement 1—source data 1.** PLCE1 promotes cell migration and invasion in ECA-109 cells.

**Figure supplement 2.** PLCE1 silencing inhibits cell migration and invasion in ECA-109 cells.

**Figure supplement 2—source data 1.** PLCE1 silencing inhibits cell migration and invasion in ECA-109 cells.

**Figure supplement 3.** Activation of PLCE1 stimulates cell migration and invasion in ECA-109 cells.

**Figure supplement 3—source data 1.** Activation of PLCE1 stimulates cell migration and invasion in ECA-109 cells.

**Figure supplement 4.** Inhibition of PLCE1 reduces cell migration and invasion in ECA-109 cells.

**Figure supplement 4—source data 1.** Inhibition of PLCE1 reduces cell migration and invasion in ECA-109 cells.

receptor (IP3R) inhibitor, and found that 2-APB treatment almost completely reversed PLCE1-induced the intracellular $Ca^{2+}$ (*Figure 5—figure supplement 1*). Accordingly, PLCE1-induced cell migration and invasion were replenished by 2-APB in ECA-109 cells (*Figure 5—figure supplement 2A–D*). The changes in EMT gene expression induced by PLCE1 were largely reversed by 2-APB treatment (*Figure 5—figure supplement 2E*). Similar to ECA-109 cells, 2-APB induced phenotypes in KYSE-150 and TE-1 cells (*Figure 5—figure supplement 3*; *Figure 5—figure supplement 4*). BAPTA-AM, an intracellular $Ca^{2+}$ chelator, dampened PLCE1-induced cell migration and invasion in ECA-109 cells (*Figure 5—figure supplement 5*). PKC is a promising downstream target of intracellular $Ca^{2+}$ (*Kadamur and Ross, 2013*). Therefore, we investigated whether the PLCE1-induced cell growth was dependent on PKC activation. To this end, midostaurin was used to inhibit PKC, and our data showed that midostaurin almost completely abolished cell migration and invasion induced by PLCE1 (*Figure 5—figure supplement 6*).

It has been shown that PKC positively regulates the expression and stability of β-Catenin in a GSK-3β-dependent manner (*Duong et al., 2017*; *Gwak et al., 2006*; *Liu et al., 2018*; *Ryu and Han, 2015*; *Tejeda-Muñoz et al., 2015*). Of note, β-Catenin is considered as a positive regulator in the EMT process (*Valenta et al., 2012*). Therefore, we predicted that PLCE1 stimulates cell migration and invasion via the axis of PKC/GSK-3β/β-Catenin. To test this prediction, ECA-109 cells were treated with an IP3R inhibitor (2-APB), $Ca^{2+}$ chelator (BAPTA-AM), or PKC inhibitor (midostaurin), all of which successfully inhibited PKC activity, as evidenced by reduced phosphorylated PKC (p-PKC) (*Figure 5A–C*; *Figure 5—figure supplement 7A–C*). As a result, phosphorylated GSK-3b(p-GSK3b) was decreased by 2-APB, BAPTA-AM, and Midostaurin (*Figure 5A–C*; *Figure 5—figure supplement 7A–C*), indicating GSK-3β kinase activity was upregulated. Accordingly, phosphorylated β-Catenin (p-β-Catenin) was increased, leading to degradation of β-Catenin (*Figure 5A-C*; *Figure 5—figure supplement 7A–C*). As a downstream target of β-Catenin, MMP2 expression was inhibited by all three tested chemicals (*Figure 5A–C*; *Figure 5—figure supplement 7A–C*). The immunofluorescence data also showed that PLCE1-induced β-Catenin expression could be counteracted by 2-APB, BAPTA-AM, and Midostaurin (*Figure 5D*). These results clearly indicate that PLCE1-induced cell migration and invasion in ESCC are likely via the axis of PKC/GSK-3β/β-Catenin. Furthermore, we observed that PLCE1-induced signal transduction in the axis of PKC/GSK-3β/β-Catenin could be inhibited by TAK1 (*Figure 5E*; *Figure 5—figure supplement 7D*). The expression of β-Catenin induced by PLCE1 was largely reversed by TAK1 (*Figure 5F*). However, inactive TAK1 (TAK1 K63W) showed no such effect (*Figure 5G*; *Figure 5—figure supplement 7E*). Moreover, TAK1 failed to affect the PLCE1 S1060A induced signal cascade in the PKC/GSK-3β/β-Catenin axis (*Figure 5H*; *Figure 5—figure supplement 7F*). This further indicates that TAK1 phosphorylates PLCE1 at S1060 to inhibit its activity and downstream signal transduction in the PKC/GSK-3β/β-Catenin axis.

## TAK1 inhibition promotes ESCC metastasis in vivo

To examine TAK1-regulated ESCC metastasis in vivo, a xenograft model with nude mice was used. Mice were injected with ECA-109 cells via the tail vein, and treated daily with Takinib (50 mg/kg) or corn oil (vehicle) for consecutive 15 days. Eight weeks later, the mice were sacrificed for analysis of cancer cell metastasis. We found that four of six mice in the Takinib group developed lung metastasis, while only one of six mice in the vehicle group developed lung metastasis. Moreover, the number of metastatic nodules in lung tissues was higher in Takinib-treated mice (*Figure 6A–C*). In addition, the effects of TAK1 inhibition on the axis of PKC/GSK-3β/β-Catenin was examined. As shown in *Figure 6D*,

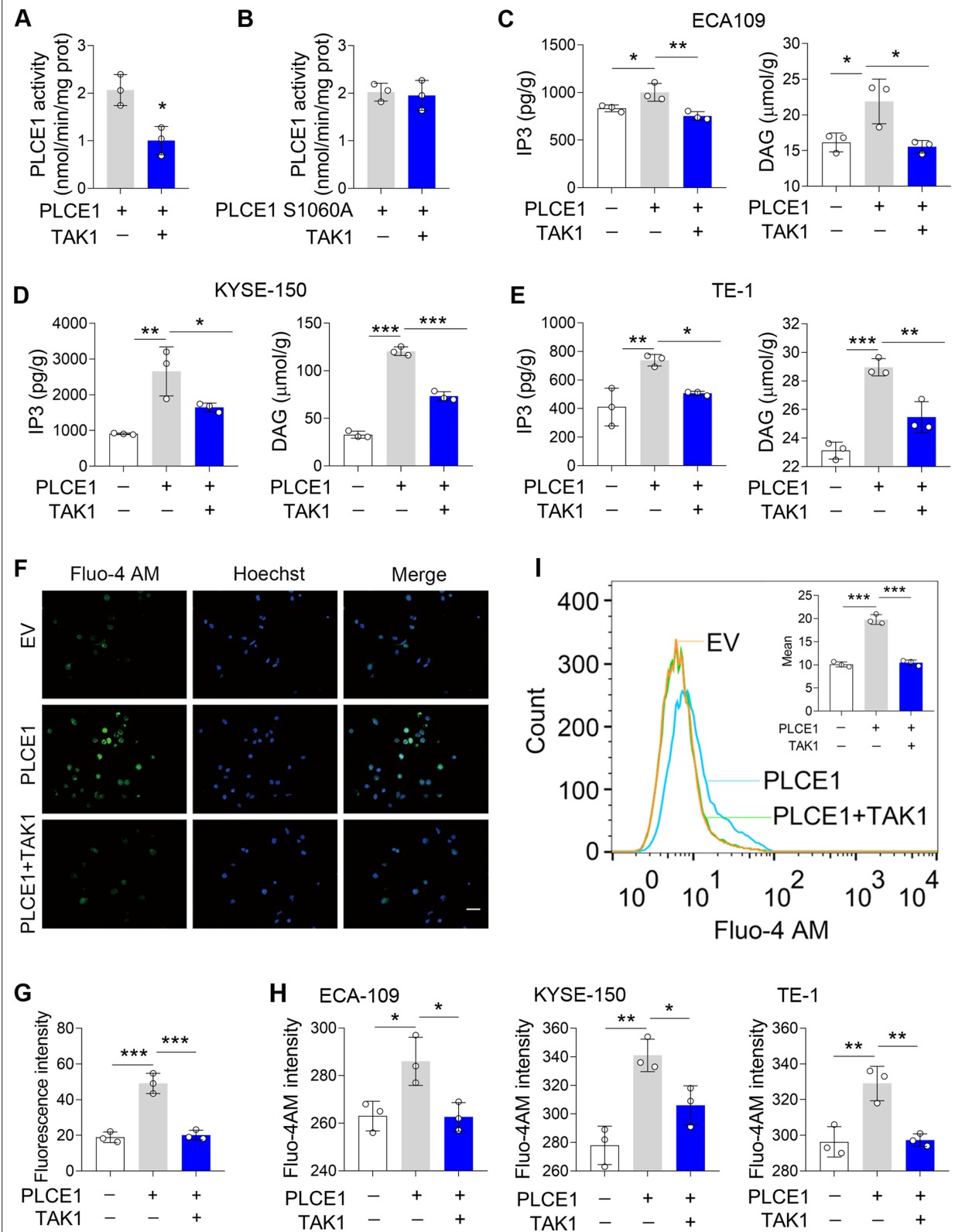

**Figure 4.** TAK1 inhibits PLCE1 enzyme activity. (**A–B**) Effects of TAK1 on PLCE1 (**A**) and PLCE1 S1060A (**B**) enzyme activity. ECA-109 cells were co-transfected with the plasmids expressing *Plce1-Myc* or *Map3k7*. 24 hr post-transfection, cells were subjected to pull-down assay by using the beads with anti-Myc antibody. PLCE1 enzyme activity was assayed by Phospholipases C (PLC) Activity Assay Kit. n=3 biologically independent replicates. (**C–E**) TAK1 abolishes PLCE1-induced inositol 1,4,5-trisphosphate (IP3) and diacylglycerol (DAG) in ECA-109 (**C**), KYSE-150 (**D**), and TE-1 cells (**E**). Cells

*Figure 4 continued on next page*

*Figure 4 continued*

were transfected with the plasmids bearing *Plce1* or *Map3k7* as indicated. 24 hr post-transfection, cells were harvested for measuring IP3 and DAG. n=3 biologically independent replicates. (**F**) TAK1 attenuates PLCE1-induced intracellular $Ca^{2+}$ ($[Ca^{2+}]$). ECA-109 cells were transfected with the plasmids bearing *Plce1* or *Map3k7* as indicated. $[Ca^{2+}]$ was labeled with Fluo-4 AM, which was then detected by a fluorescent microscope. Scale bar = 20 μm. (**G**) Quantified fluorescence intensity of $[Ca^{2+}]$ in ECA-109 cells. n=3 biologically independent replicates. (**H**) Fluorescence intensity of Fluo-4 in ECA-109, KYSE-150, and TE-1 cells was examined with a fluorospectrophotometer. n=3 biologically independent replicates. (**I**) Flow cytometry analysis of $[Ca^{2+}]$. Cell treatments were described in (**F**). Data are presented as mean ± SD. Statistical significance was tested by unpaired Student's t-test (**A–B**) or two-tailed one-way ANOVA test (**C–E, G–I**). *$p<0.05$, **$p<0.01$, and ***$p<0.001$.

The online version of this article includes the following source data and figure supplement(s) for figure 4:

**Source data 1.** TAK1 inhibits PLCE1 enzyme activity.

**Figure supplement 1.** The negative impact of TAK1 on PLCE1-induced cell migration and invasion requires TAK1 kinase activity.

**Figure supplement 1—source data 1.** The negative impact of TAK1 on PLCE1-induced cell migration and invasion requires TAK1 kinase activity.

**Figure supplement 2.** Effects of TAK1 on p-IKK, p-JNK, p-ERK, and p-P38 MAPK in ECA-109 cells.

**Figure supplement 2—source data 1.** Effects of TAK1 on p-IKK, p-JNK, p-ERK, and p-P38 MAPK in ECA-109 cells.

**Figure supplement 2—source data 2.** PDF file containing original western blots for *Figure 4—figure supplement 2A and C*, indicating the relevant bands.

**Figure supplement 2—source data 3.** Original files for western blot analysis displayed in *Figure 4—figure supplement 2A and C*.

TAK1 inhibition by Takinib activated PKC, leading to increased p-GSK-3β levels. As a result, p-β-Catenin was reduced and MMP2 expression was upregulated (*Figure 6D and E*). Notably, the p-TAK1 levels were decreased by Takinib, indicating that TAK1 activity was successfully inhibited (*Figure 6D and E*). Accordingly, p-PLCE1 expression was decreased in Takinib-treated mice (*Figure 6D and E*). The body weight growth rate was not affected by Takinib (*Figure 6—figure supplement 1A*). These data indicate that TAK1 represses ESCC metastasis in vivo, and this benefit is likely due to TAK1-mediated phosphorylation of PLCE1 at S1060.

## PLCE1 facilitates ESCC metastasis in vivo

To further verify the function of PLCE1 in ESCC metastasis, we generated a stable cell line with PLCE1 knockdown using a lentivirus bearing PLCE1 shRNA (LV-shPLCE1) (SI Appendix, *Figure 6—figure supplement 1B and C*). Cells with low PLCE1 expression were injected into nude mice via the tail vein to generate a mouse xenograft tumor model. Eight weeks later, the mice were sacrificed, and the nodule number and incidence rate were reduced in mice injected with LV-shPLCE1 cells (*Figure 7A–C*). In addition, western blot analysis showed that PLCE1 silencing inhibited signal transduction in the PKC/GSK-3β/β-Catenin axis, as evidenced by reduced p-PKC and p-GSK-3β, and increased p-β-Catenin (*Figure 7D and E*). Accordingly, MMP2 was decreased (*Figure 7D and E*). These results further confirm that PLCE1 plays a key role in cancer cell metastasis in ESCC.

## Discussion

TAK1 is a serine/threonine kinase and a major member of the MAPK family (*Zhu et al., 2021*). In response to various cytokines, pathogens, lipopolysaccharides, hypoxia, and DNA damage, the E3 ligases TRAF2 and TRAF6 activate TAK1 through Lys63-linked polyubiquitylation, and then TAK1 undergoes auto-phosphorylation at Ser192 and Thr184/187 to achieve full activation (*Skaug et al., 2009*; *Sorrentino et al., 2008*). Consequently, activated TAK1, in turn, phosphorylates downstream substrates in order to initiate the NF-kB and MAPKs (i.e. ERK, p38 MAPK, and JNK) signaling pathways, thereby participates in cellular inflammation, immune response, fibrosis, cell death, and cancer cell invasion and metastasis (*Yang et al., 2022*; *Zhou et al., 2021*). TAK1 has been shown to decrease in high-grade human prostate cancer, and TAK1 deficiency promotes prostate tumorigenesis by increasing androgen receptor (AR) protein levels and activity or by activating the p38 MAPK pathway (*Huang et al., 2021*; *Wu et al., 2012*). Similarly, hepatocyte-specific TAK1 ablation drives RIPK1 kinase-dependent inflammation to promote liver fibrosis and HCC (*Su et al., 2023*; *Tan et al., 2020*; *Xia et al., 2021*). Furthermore, TAK1 represses the transcription of human telomerase and activates the tumor suppressor protein LKB1, indicating that TAK1 is a tumor suppressor (*Adhikari et al., 2007*; *Fujiki et al., 2007*; *Xie et al., 2006*). Consistent with these findings, in our previous study, we

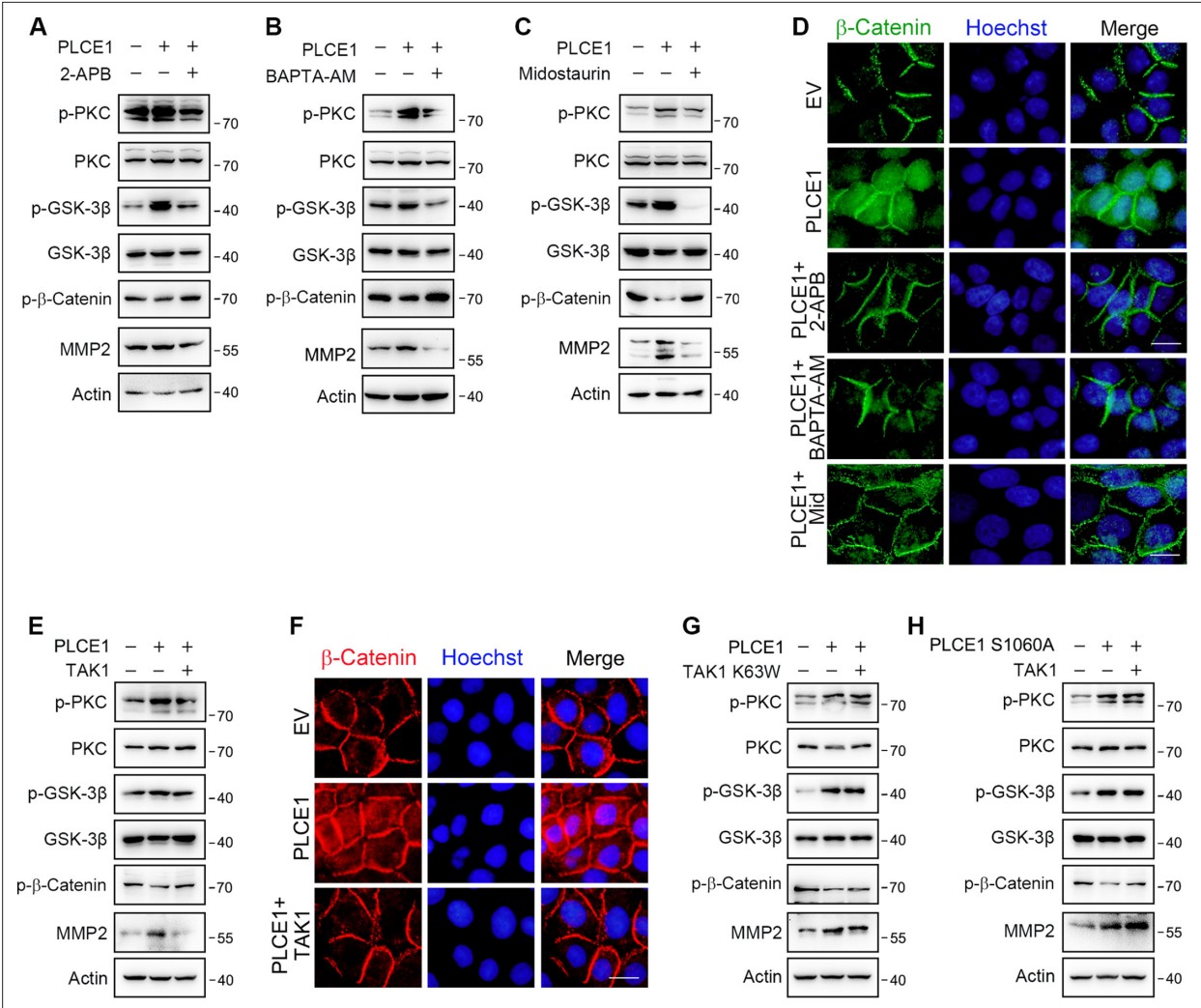

**Figure 5.** TAK1 inhibits PLCE1-induced signal transduction in the axis of PKC/GSK-3β/β-Catenin. (**A**) IP3R blocking inhibits PLCE1-induced signal transduction in the axis of PKC/GSK-3β/β-Catenin. ECA-109 cells were transfected with the plasmid expressing *Plce1* for 6 hr and then treated with 2-APB (10 μM) for additional 18 hr. (**B**) [Ca²⁺] blocking represses signal transduction in the axis of PKC/GSK-3β/β-Catenin induced by PLCE1. ECA-109 cells were transfected with the plasmid expressing *Plce1* for 6 hr and then treated with BAPTA-AM (10 μM) for additional 18 hr. (**C**) PKC inhibition blocks PLCE1 stimulated signal transduction in the axis of PKC/GSK-3β/β-Catenin. ECA-109 cells were transfected with the plasmid expressing *Plce1*. 6 hr post-transfection, cells were treated with 100 nM of Midostaurin for additional 18 hr. (**D**) PKC inhibition represses PLCE1-induced nuclear translocation of β-Catenin in ECA-109 cells. Cells were transfected with the plasmid expressing PLCE1. 6 hr post-transfection, 2-APB (10 μM), BAPTA-AM (10 μM), or Midostaurin (100 nM) was added in culture medium, and cells were cultured for additional 18 hr. Scale bar = 10 μm. Immunofluorescence was used to examine subcellular distribution of β-Catenin. (**E**) TAK1 counteracts PLCE1-induced signal transduction in the axis of PKC/GSK-3β/β-Catenin. ECA-109 cells were transfected with the plasmids expressing *Plce1* or *Map3k7* as indicated for 24 hr. (**F**) TAK1 reduces PLCE1-induced nuclear distribution of β-Catenin in ECA-109 cells. Cells were transfected with the plasmids expressing *Plce1* or *Map3k7* as indicated. Scale bar = 10 μm. (**G**) Dominant negative TAK1 (K63W) fails to block signal transduction in the axis of PKC/GSK-3β/β-Catenin/MMP2 induced by PLCE1. ECA-109 cells were transfected with the plasmids expressing *Plce1* or mutated *Map3k7* (TAK1 K63W) for 24 hr. (**H**) TAK1 has no effect on PLCE1 S1060A-induced signal transduction in the axis of PKC/GSK-3β/β-Catenin. ECA-109 cells were transfected with the plasmids expressing PLCE1 S1060A or TAK1 for 24 hr. Protein levels were analyzed by western blot, and Actin was used as a loading control. Representative blots were shown.

The online version of this article includes the following source data and figure supplement(s) for figure 5:

**Source data 1.** PDF file containing original western blots for *Figure 5A, B, C, E, G, and H*, indicating the relevant bands.

**Source data 2.** Original files for western blot analysis displayed in *Figure 5A, B, C, E,G, and H*.

**Figure supplement 1.** IP3R blockade inhibits PLCE1-induced intracellular calcium accumulation.

**Figure supplement 1—source data 1.** IP3R blockade inhibits PLCE1-induced intracellular calcium accumulation.

**Figure supplement 2.** 2-APB treatment counteracts PLCE1-induced cell migration.

*Figure 5 continued on next page*

*Figure 5 continued*

**Figure supplement 2—source data 1.** 2-APB treatment counteracts PLCE1-induced cell migration.

**Figure supplement 3.** IP3R inhibition represses PLCE1-stimulated cell migration and invasion in KYSE-150 cells.

**Figure supplement 3—source data 1.** IP3R inhibition represses PLCE1-stimulated cell migration and invasion in KYSE-150 cells.

**Figure supplement 4.** IP3R inhibition reduces PLCE1-stimulated cell migration and invasion in TE-1 cells.

**Figure supplement 4—source data 1.** IP3R inhibition reduces PLCE1-stimulated cell migration and invasion in TE-1 cells.

**Figure supplement 5.** [$Ca^{2+}$] blockade inhibits PLCE1-induced cell migration and invasion in ECA-109 cells.

**Figure supplement 5—source data 1.** [$Ca^{2+}$] blockade inhibits PLCE1-induced cell migration and invasion in ECA-109 cells.

**Figure supplement 6.** PKC inhibition counteracts PLCE1-induced cell migration and invasion.

**Figure supplement 6—source data 1.** PKC inhibition counteracts PLCE1-induced cell migration and invasion.

**Figure supplement 7.** TAK1 mitigates PLCE1-induced signal transduction in the axis of PKC/GSK-3β/β-Catenin.

**Figure supplement 7—source data 1.** TAK1 mitigates PLCE1-induced signal transduction in the axis of PKC/GSK-3β/β-Catenin.

found that TAK1 expression was reduced in esophageal squamous tumors when compared to that in adjacent normal tissues, and that TAK1 was negatively correlated with esophageal squamous tumor patient survival (*Shi et al., 2021*). In this study, we extended our research on TAK1 and observed that TAK1 inhibits cell migration and invasion in ESCC. In contrast, TAK1 plays a positive role in tumor cell proliferation, migration, invasion, colony formation, and metastasis, especially in breast, pancreatic, and non-small-cell lung cancers (*Kim et al., 2023*; *Santoro et al., 2020*; *Tripathi et al., 2019*). Therefore, TAK1 may play a pleiotropic role in various cancer cell types.

As a kinase, TAK1 is considered as a central regulator of tumor cell proliferation, migration, and invasion and has been demonstrated to play tumor suppression or activation role via the NF-κB and MAPK activation (*Mukhopadhyay and Lee, 2020*; *Wang et al., 2021*; *Zhang et al., 2023*). However, the precise molecular and cellular mechanisms through which TAK1 regulates ESCC metastasis remain unclear. As previously reported, using co-immunoprecipitation coupled with mass spectrometry (MS/MS), we identified RASSF9 as a downstream target of TAK1 to repress on cell proliferation in ESCC (*Shi et al., 2021*). By phosphorylating RASSF9 at S284, TAK1 negatively regulates ESCC proliferation (*Shi et al., 2021*). We also found that PLCE1 was present in the immunocomplex and was phosphorylated at S1060 in the presence of TAK1 in ECA-109 cells. It should be mentioned that, TAK1 inhibitors cannot completely abolish p-PLCE1 S1060 in cells and mice, which might be due to some other kinases also targeting PLCE1 at S1060. As a member of the human phosphoinositide-specific PLC family, PLCE1 is activated by various intracellular and extracellular signaling molecules, including hormones, cytokines, neurotransmitters, and growth factors (*Kadamur and Ross, 2013*). Numerous studies have shown that PLCE1 plays a key role in cancer development and progression via various pathways (*Abnet et al., 2010*; *Fan et al., 2019*; *Ghosh et al., 2021*; *He et al., 2016*; *Yue et al., 2019*). Moreover, PLCE1 has been identified as a susceptibility gene for ESCC (*Abnet et al., 2010*; *Wang et al., 2010*). This suggests that PLCE1 is a potential downstream target for transducing the negative effects of TAK1 on ESCC migration and invasion.

Once activated, PLCE1 catalyzes the hydrolysis of PIP2 on the cell membrane in order to produce two secondary messengers IP3 and DAG; IP3 induces $Ca^{2+}$ release from the ER into the cytoplasm via IP3R; both $Ca^{2+}$ and DAG are potent activators of PKC. PLCE1 regulates cell growth, proliferation, and differentiation, thereby playing a key role in tumor growth and development (*Bunney and Katan, 2010*; *Kadamur and Ross, 2013*; *Land and Rubin, 2017*; *Smrcka et al., 2012*). Consistent with these observations, we observed that PLCE1-overexpressing cells exhibited a higher production of intracellular IP3, DAG, and intracellular $Ca^{2+}$. We found that PLCE1 overexpression accelerated cell migration and invasion, while PLCE1 knockdown resulted in the opposite phenotype. Moreover, in the presence of TAK1, PLCE1-induced cell migration and invasion were largely counteracted. Considering that TAK1 is a protein kinase, we propose that TAK1 phosphorylates PLCE1 at S1060 to inhibit its enzymatic activity, thus impedes cell migration and invasion in ESCC. Indeed, PLCE1 activity was blunted in the presence of TAK1, whereas the mutated PLCE1 (PLCE1 S1060A) was not affected. It has been shown that PKC positively regulates the expression and stability of β-Catenin in a GSK-3β-dependent manner (*Duong et al., 2017*; *Gwak et al., 2006*; *Ryu and Han, 2015*). Therefore, we investigated the impact of PLCE1 on the PKC/GSK-3β/β-Catenin axis. As expected, our data revealed

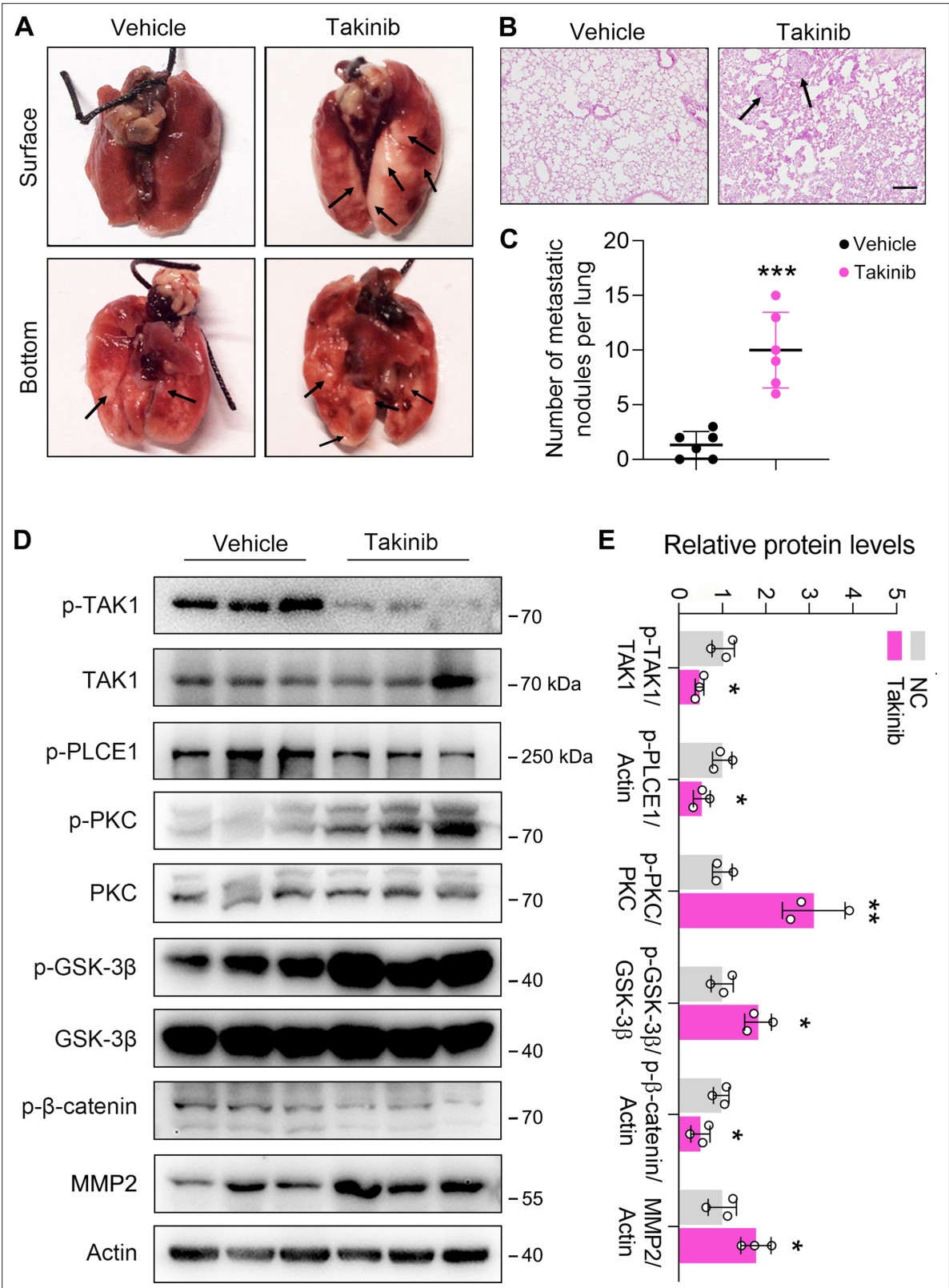

**Figure 6.** Inhibition of TAK1 by Takinib promotes esophageal squamous cell carcinoma (ESCC) metastasis in nude mice. Each mouse was intravenously injected with 1×10⁶ ECA-109 cells diluted in 100 µl PBS. Mice were treated with Takinib at the dosage of 50 mg/kg/day for 15 days, mice in control group were received vehicle (corn oil). Eight weeks later, mice were sacrificed, and the lungs and livers from each group were collected and photographed. (**A**) Typical images of specimens. (**B**) Hematoxylin and eosin staining of metastatic nodules in lungs. (**C**) The number of nodules in lungs. (**D**) Takinib

*Figure 6 continued on next page*

*Figure 6 continued*

treatment induces signal transduction in the axis of PKC/GSK-3β/β-Catenin. Protein levels were analyzed by western blot, and Actin was used as a loading control. n=3 biologically independent replicates. (E) Quantitative analysis of the western blot data shown in (D). Data are presented as mean ± SD. Statistical significance was tested by unpaired Student's t-test. *p<0.05, **p<0.01, and ***p<0.001.

The online version of this article includes the following source data and figure supplement(s) for figure 6:

**Source data 1.** Inhibition of TAK1 by Takinib promotes esophageal squamous cell carcinoma (ESCC) metastasis in nude mice.

**Source data 2.** PDF file containing original western blots for *Figure 6D*, indicating the relevant bands.

**Source data 3.** Original files for western blot analysis displayed in *Figure 6D*.

**Figure supplement 1.** The effects of Takinib on mouse body weight and knockdown of PLCE1 by lentivirus in ECA-109 cells.

**Figure supplement 1—source data 1.** The effects of Takinib on mouse body weight and knockdown of PLCE1 by lentivirus in ECA-109 cells.

**Figure supplement 1—source data 2.** PDF file containing original western blots for *Figure 6—figure supplement 1C*, indicating the relevant bands.

**Figure supplement 1—source data 3.** Original files for western blot analysis displayed in *Figure 6—figure supplement 1C*.

that PLCE1 promotes cell migration and invasion in ESCC by activating the PKC/GSK-3β/β-Catenin pathway, results in a series of phosphorylation modification in PKC and GSK-3β. As a result, phosphorylated β-Catenin decreased, leading to a higher stability of β-Catenin, which eventually promotes the EMT process by affecting epithelial and mesenchymal gene expression.

In summary, our findings have indicated TAK1 negatively regulates the cell migration and invasion of ESCC. Mechanistically, TAK1 phosphorylates PLCE1 at residue S1060 to inhibit phospholipase activity, leading to a reduction in IP3 and DAG. Consequently, PKC activity was blunted, which results in decreased phosphorylated GSK-3β and increased phosphorylated β-Catenin. As a consequence, the stability of β-Catenin was decreased and its transcriptional activity was blunted. Thus, epithelial marker gene expression was upregulated by TAK1, whereas the mesenchymal marker gene expression was downregulated. These outcomes eventually lead to a breakdown in the cell migration and invasion of ESCC. Hence, our data revealed a new facet of TAK1 in the EMT process in ESCC by inhibiting PLCE1 activity and its downstream signal transduction in the axis of PKC/GSK-3β/β-Catenin. Moreover, TAK1 and, together with its downstream target, PLCE1, are potential drug targets for the development of agents for treating ESCC.

## Methods
### Antibodies, inhibitors, and growth factors

The following antibodies were used: rabbit polyclonal anti-PLCE1 (#PA5-100856; Invitrogen, Carlsbad, CA, USA). Rabbit monoclonal antibodies against TAK1 (#5206), phospho-TAK1 (Ser412, #9339), PKCα (#2056), phospho-PKC (pan) (gamma Thr514) (#38938), GSK-3β (#9315), phospho-GSK-3β (Ser9) (#5558), β-Catenin (#8480), phospho-β-Catenin (Ser33/37/Thr41) (#9561), MMP-2 (#87809), mouse monoclonal antibodies against Actin (#3700), Myc-Tag (Sepharose Bead Conjugate) (#55464), Epithelial-Mesenchymal Transition (EMT) antibody sample kit (#9782), phosphor-IKK (#2078), IKK (#61294), phosphor-JNK (#4668), JNK (#9252), phospho-ERK (#4370), ERK (#9102), phospho-P38 MAPK (#9211), P38 MPAK (#9212), and normal rabbit IgG (#2729) were from Cell Signaling Technology (Beverley, MA, USA). TAK1 inhibitors 5Z-7-oxozeaenol (O9890), NG25 (SML1332), and Takinib (SML2216) (#662009) were purchased from Sigma-Aldrich (St. Louis, MO, USA). An intracellular calcium chelator BAPTA-AM (HY-100545), a PKC inhibitor Midostaurin (HY-10230), and an IP3R antagonist (HY-W009724) were obtained from MedChemExpress (NJ, USA). EGF Recombinant Human Protein (#PHG0311) was purchased from Gibco (Carlsbad, CA, USA).

### Cell culture

Human ESCC cell lines ECA-109, KYSE-150, and TE-1 were purchased from the Shanghai Institute of Biochemistry and Cell Biology (Shanghai, China). HEK-293 cells were purchased from American Type Culture Collection (ATCC, Manassas, VA, USA). All cells were cultured in Dulbecco's modified Eagle's medium (DMEM) (Hyclone, UT, USA) containing 10% fetal bovine serum (FBS, Gibco, Carlsbad, CA), 100 U/ml penicillin, and 100 µg/ml streptomycin (Life Technologies, Carlsbad, CA, USA) at 37°C in

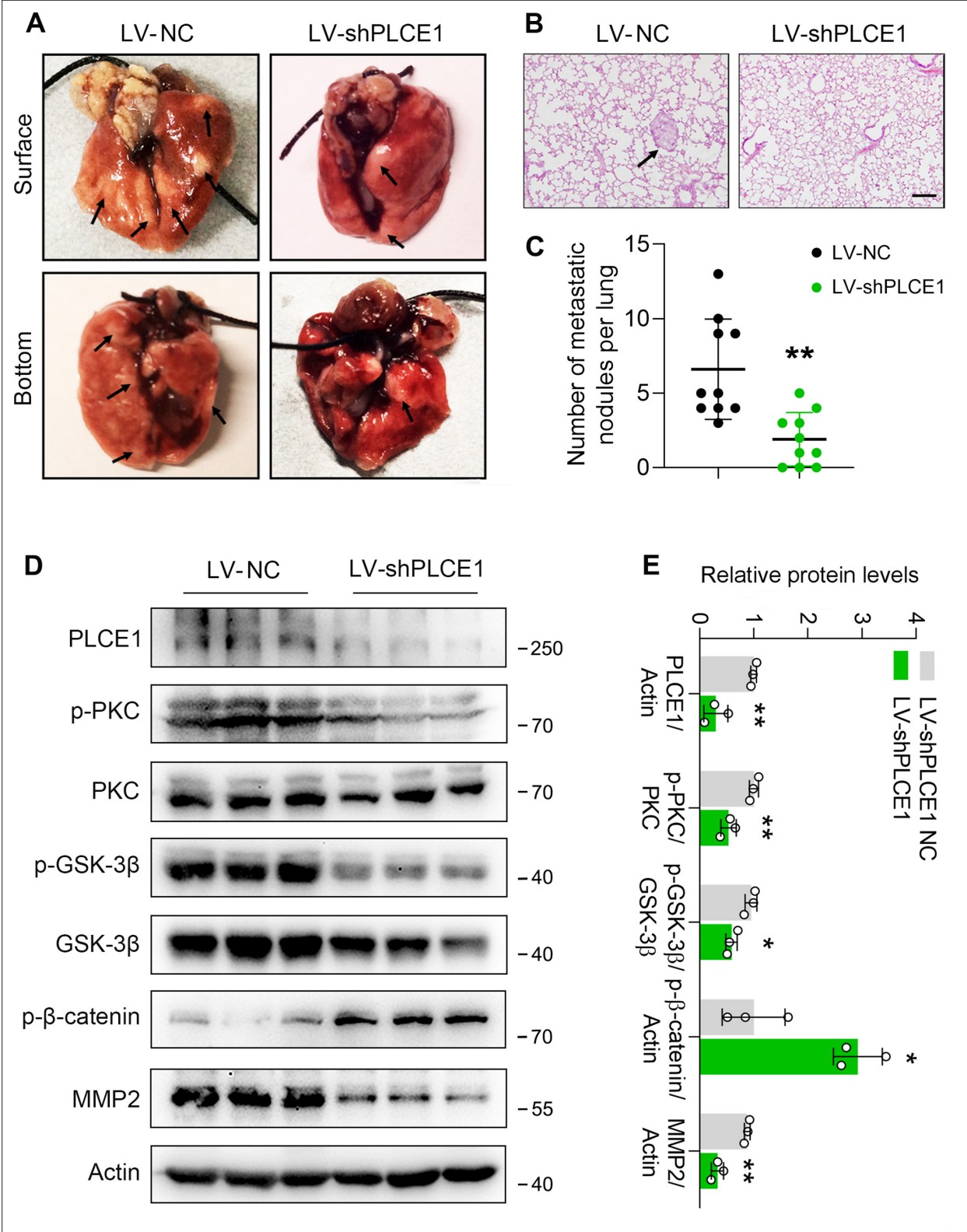

**Figure 7.** PLCE1 knockdown inhibits esophageal squamous cell carcinoma (ESCC) metastasis in nude mice. ECA-109 cells were transduced with lentivirus bearing PLCE1 shRNA (LV-shPLCE1) or NC shRNA (LV-shPLCE1 NC). Each mouse was intravenously injected with the LV transduced cells (1×10[6] cells/mouse). Eight weeks later, mice were sacrificed, and the lungs and livers from each group were collected and photographed. (**A**) Typical images of lung specimens. (**B**) Hematoxylin and eosin staining of metastatic nodules in lungs. (**C**) The number of nodules in lungs. (**D**) PLCE1 knockdown

*Figure 7 continued on next page*

*Figure 7 continued*

represses signal transduction in the axis of PKC/GSK-3β/β-Catenin. Protein levels were analyzed by western blot, and Actin was used as a loading control. n=3 biologically independent replicates. (**E**) Quantitative analysis of the western blot data shown in (**D**). Data are presented as mean ± SD. Statistical significance was tested by unpaired Student's t-test. **p<0.01 and ***p<0.001.

The online version of this article includes the following source data for figure 7:

**Source data 1.** PLCE1 knockdown inhibits esophageal squamous cell carcinoma (ESCC) metastasis in nude mice.

**Source data 2.** PDF file containing original western blots for *Figure 7D*, indicating the relevant bands.

**Source data 3.** Original files for western blot analysis displayed in *Figure 7D*.

a humidified atmosphere of 5% $CO_2$. All cultures were routinely screened for the absence of mycoplasma contamination using the ATCC Universal Mycoplasma Detection Kit (#30-1012K).

## Human esophageal squamous tumor specimens

This study utilized pathological specimens obtained from 10 patients with histologically confirmed invasive ESCC at the Affiliated Hospital of Nantong University. All specimens were collected after obtaining written informed consent from participants, in strict accordance with the ethical principles outlined in the Declaration of Helsinki. The cohort consisted of seven male and three female patients, aged 43–70 years. The research protocol received ethical approval from the Institutional Review Board of the Affiliated Hospital of Nantong University (Ethical Approval No.: 2015 L132).

## Generation of antibodies against phospho-PLCE1 (S1060)

A rabbit polyclonal antibody that recognizing phospho-PLCE1 (S1060) was raised against the c-WSARNPS(p)PGTSAK peptide at Absin (Shanghai, China). Briefly, the phosphorylated polypeptide c-WSARNPS(p)PGTSAK was synthesized as the target peptide. Two rabbits were then immunized with the target peptide antigen. The non-phosphorylated polypeptide c-WSARNPSPGTSAK was used as a control. The rabbits were sacrificed and blood was collected for further purification. The prepared antibody was verified and stored at –20°C until use.

## CRISPR-Cas9-mediated TAK1 deletion

*Map3k7* was deleted using the CRISPR-Cas9 method described previously (*Shi et al., 2021*). Briefly, we designed an integrated vector based on OriP/EBNA1 (epiCRISPR) bearing gRNA, Cas9, and puromycin resistance genes. ECA-109 cells were transfected with plasmids using Lipofectamine 2000 (Invitrogen), according to the manufacturer's instructions. The cells were cultured in a medium containing 2.5 µg/ml puromycin for 1 week. Surviving stable cells were used for further experiments. The gRNA sequences are listed in *Supplementary file 1*.

## Plasmid construction

The full-length coding regions of *Map3k7* was synthesized by Heyuan Biotechnology Company (Shanghai, China). The synthesized *Map3k7* was cloned into pcDNA3.1(+) (Invitrogen, Carlsbad, CA, USA) vector using EcoR I and Xho I. Correct construction was confirmed by DNA sequencing. The plasmid of pRK5-N-myc PLCE1 was a gift from Dr. Friedhelm Hildebrandt (*Chaib et al., 2008*). PLCE1 (S1060A) was generated using a PCR-based mutagenesis kit (Stratagene, La Jolla, CA, USA) using pRK5-N-myc-human PLCE1 as template. The primer sequences are listed in *Supplementary file 1*. All plasmids were confirmed by sequencing. For transient transfection, the plasmids were transfected into ESCC cells with Lipofectamine 2000 (Invitrogen, Carlsbad, CA, USA), according to the manufacturer's instructions.

*Map3k7* short interfering RNAs (siRNAs), *Plce1* siRNAs, and a corresponding scrambled siRNA were chemically synthesized at RiboBio (Guangzhou, China). The siRNA sequences are listed in *Supplementary file 1*. The siRNAs were transfected into ECA-109 cells with Lipofectamine RNAiMAX (Invitrogen, Carlsbad, CA, USA), in accordance with the instructions.

## Lentivirus transduction

LV-*Map3k7* shRNA and LV-NC shRNA were produced at Hanbio (Shanghai, China). LV-*Plce1* shRNA (LV-shPLCE1) and LV-NC shRNA were produced at OBiO (Shanghai, China). Lentivirus transduction

was performed as previously described (*Shi et al., 2021*). To generate a stale cell line with low PLCE1 expression, ECA-109 cells were transduced with LV-shPLCE1. 24 hr post-transduction, cells were treated with 2.5 µg/ml puromycin to eliminate non-transduced cells. The efficiency of lentivirus with knockdown of TAK1 or PLCE1 were verified by qRT-PCR and western blot analysis.

## Quantitative real time-PCR

Total RNA was isolated from the ESCC cells using TRIzol Reagent (Invitrogen, Carlsbad, CA, USA) and quantified. Then, reversely transcribed into First strand cDNA using the PrimeScript RT reagent kit (Takara, Tokyo, Japan) according to the manufacturer's instruction. qRT-PCR was used to analyze the expression level of target gene mRNA by using a Fast Start Master SYBR Green Kit (Roche) on the RocheLightCycler96 instrument and software (Roche, Basel, Switzerland). The qRT-PCR conditions were as follows: 95°C for 6 min, followed by 40 cycles of 95°C for 10 s, 60°C for 30 s, 72°C for 10 s. Primer sequences were shown in *Supplementary file 2*. Relative quantitation of target genes was normalized to that of GAPDH mRNA and were analyzed using the $2^{-\triangle\triangle CT}$ method.

## Protein extraction and western blot analysis

Western blotting was performed as previously described (*Tan et al., 2022*). Briefly, total protein was extracted from the cells and tissues using protein lysis buffer supplemented with protease and phosphatase inhibitors (Roche Applied Science, Penzberg, Germany). Protein concentrations were assayed by the Pierce BCA Protein Assay Kit (Thermo Scientific, Waltham, MA, USA). Subsequently, equal amounts of protein were separated with 6% or 10% SDS-PAGE, and then protein were transferred onto 0.45 µm PVDF membranes. After blocking with 5% BSA in TBST for 1 hr, membranes were incubated with primary antibodies at 4°C overnight. The membranes were then incubated with horseradish peroxidase-conjugated secondary antibodies for 1 hr at room temperature. Protein signals were visualized using an enhanced chemiluminescence (ECL) reagent (Thermo Fisher Scientific, Waltham, MA, USA) and quantitatively analyzed using ImageJ (NIH). Actin was used as a loading control.

## Wound healing assay

For wound healing assay, ibidi Culture-Inserts (ibidi GmbH) were used to guarantee the uniformity of each initial (0 hr) wound. In brief, $3\times10^4$ cells were seeded into the ibidi Culture-Inserts (70 µl cell suspension for each side of the scratch chamber) and incubated in medium containing 10% FBS. After 24 hr, cells were grown to almost 100% confluence. Then, the scratch chambers were removed to make the wounds and rinsed with PBS for three times to remove the suspended cells. Cells were continually cultured in DMEM supplemented with 1% FBS and photographed randomly at 0 hr, 24 hr, and 48 hr, respectively. Wound closure was calculated using the ImageJ software according to the following formula: Wound healing rate (%) = (24 hr or 48 hr wound area - 0 hr wound area)/0 hr wound area × 100%.

## Transwell assay

Cell migration and invasion were determined using transwell assay. For migration assay, $5\times10^4$ cells in 200 µl of serum-free culture medium were placed into the upper chambers (8.0 µm pore size, Corning, NY, USA) in 24-well plates. For cell invasion assay, $5\times10^4$ cells in 200 µl of serum-free culture medium were placed into the upper chambers coated with 80 µl of Matrigel (BD Bioscience). The lower chamber was added with 500 µl of DMEM supplemented with 10% FBS. After incubating for 24 hr at 37°C, the non-migrating cells in the upper chamber were gently wiped with a cotton swab. The chambers were then fixed with 4% paraformaldehyde for 20 min and dyed with 0.1% crystal violet for 30 min. The migration and invasive cells onto the lower side of the chamber were imaged (randomly selected five fields) and counted under a light microscope (Olympus BX51, Tokyo, Japan) at ×200.

## Immunofluorescence

Cells were seeded into a 24-well plate with glass coverslips ($2\times10^4$ per well). Next day, cells were transfected with plasmid expressing PLCE1 or TAK1 for 6 hr and then treated with BAPTA-AM (10 µM) or 2-APB (10 µM) or Midostaurin (100 nM) for additional 18 hr, respectively. For immunofluorescence staining, cells were fixed with methanol at room temperature for 15–20 min and washed three times in

PBS, each time for 10 min. Then, cells were blocked with 5% BSA solution containing 0.1–0.5% Triton X-100 for 2 hr at room temperature. After blocking, cells were incubated with rabbit anti-β-Catenin antibody overnight at 4°C, followed by an Alexa Fluor 488/594-conjugated secondary antibody incubation for 2 hr at room temperature. Finally, the nuclei were stained with Hoechst. Photographs (×400) were taken with an Olympus fluorescence microscope (BX51, Tokyo, Japan).

### In vitro kinase assay

Protein pull-down was performed using a previously reported method (2). Briefly, ECA-109 cells were transfected with a plasmid expressing Myc tagged PLCE1 (Myc-PLCE1), Myc tagged PLCE1 S1060A (Myc-PLCE1 S1060A), or S protein tagged TAK1 (SP-TAK1). 24 hr post-transfection, cells were harvested for preparing total cell lysates, which was then subjected to protein pull-down using the Myc-Tag beads (Sepharose Bead Conjugate; #55464; CST, Beverley, MA, USA) or S-protein Agarose (#69704; Millipore, Billerica, MA, USA). After 4 hr incubation on a rotator, unbound proteins were removed using ice-cold wash buffer. Myc-PLCE1, Myc-PLCE1 S1060A, and SP-TAK1 were eluted and collected. The purified SP-TAK1 was incubated with Myc-PLCE1 or Myc-PLCE1 S1060A in a kinase assay buffer (Cell Signaling Technology, #9802), in which 200 μM ATP (Cell Signaling Technology, #9804) was added. The reaction was performed at 30°C for 30 min. The reaction was terminated by adding 5×loading buffer to each sample. After boiling at 100°C for 5 min, the samples were resolved by SDS-PAGE, the resulting gels were analyzed by Coomassie blue stating or western blot.

### Measurement of intracellular-free Ca$^{2+}$

Cells were transfected with plasmid expressing PLCE1 and/or TAK1 for 6 hr and then treated with BAPTA-AM (10 μM), 2-APB (10 μM), or Midostaurin (100 nM) for additional 18 hr, respectively. Intracellular Ca$^{2+}$ levels in the treated cells were measured using a Fluo-4 AM Assay Kit (Beyotime, Jiangsu, China). The cells were then washed with PBS and incubated with 1 μM Fluo-4 AM in PBS at 37°C for 30 min. The cells were then washed with PBS three times and further incubated for 20 min to ensure that Fluo-4 AM was completely transformed into Fluo-4. Finally, Fluo-4 fluorescence intensity was detected using a confocal laser scanning fluorescence microscope (Olympus BX51, Tokyo, Japan), a fluorescence microplate (BioTek Instruments, Inc), or flow cytometry (FACSCalibur, BD Bioscience, Franklin Lakes, NJ, USA) at excitation wavelength of 488 nm and an emission wavelength of 520 nm to determine the change in intracellular Ca$^{2+}$ concentration.

### PLCE1 pull-down and phospholipase activity assay

Protein pull-down was performed using a previously reported method (*Shi et al., 2021*). HEK293 cells were transfected with plasmids expressing either PLCE1-Myc or TAK1. 24 hr post-transfection, the cells were harvested for preparing total cell lysates, which was then subjected to protein pull-down assay using the Myc-Tag beads (Sepharose Bead Conjugate; #55464; CST, Beverley, MA, USA). After 4 hr of incubation on a rotator, the unbound proteins were removed using ice-cold wash buffer. Finally, phospholipase activity bound to the beads was detected using a PLC Activity Assay Kit (Jining Shiye, Shanghai, China) according to the manufacturer's protocol. PLC catalyzes the hydrolysis of *O*-(4-nitrophenyl) choline to produce *p*-nitrophenol (PNP), which has an absorption maximum at 405 nm. The magnitude of the PLC enzyme activity was calculated by detecting the rate of increase in PNP at 405 nm.

### IP3 and DAG assays

IP3 and DAG levels in the cells were assayed using a Human Inositol 1,4,5-triphosphate enzyme-linked Immunosorbent Assay (ELISA) Kit and a Human Diacylglycerol commercial ELISA Kit (Mlbio, Shanghai, China), according to the manufacturer's instructions. The absorbance was measured at 450 nm using a microplate reader.

### Co-immunoprecipitation and MS/MS spectrometry

Co-immunoprecipitation and MS/MS assays were performed as described previously (*Shi et al., 2021*). Briefly, ECA-109 cells were transfected with a plasmid expressing *Map3k7*. At 36 hr post-transfection, cells were harvested for co-immunoprecipitation using an antibody against TAK1. The resulting immunocomplex was subjected to liquid chromatography-tandem mass spectrometry

(LC-MS/MS) (Shanghai Applied Protein Technology Co., Ltd.). MS/MS spectra were searched using the MASCOT engine (Matrix Science, London, UK; version 2.4) against UniProKB human (161584 total entries, downloaded 20180105).

## Mouse xenograft metastasis model

For in vivo metastasis assay, 4-week-old male BALB/c nude mice (18–20 g) were purchased from Shanghai Slake Laboratory Animal Co., Ltd. (Shanghai, China) and randomly divided into two groups (n=6–10 per group). Mice meeting strict inclusion criteria (age-matched males, 18–20 g, normal baseline behavior) were housed in air-filtered laminar flow cages (5 mice/cage) with a 12 hr light cycle and adequate food and water. To examine the role of PLCE1 on metastasis, each mouse was intravenously (i.v.) injected with $1 \times 10^6$ ECA-109 cells diluted in 100 µl PBS, which were transduced with LV-shPLCE1 or negative control virus. Additionally, to examine TAK1 inhibition on metastasis, each mouse was i.v. injected with $1 \times 10^6$ ECA-109 cells. The mice were then intraperitoneally injected with Takinib (50 mg/kg/day) or vehicle (corn oil) for 15 days. Blinded investigators performed all outcome assessments while separate personnel conducted treatments. Animals showing >15% weight loss or distress were excluded per protocol. The body weight of the mice was measured once per week. Eight weeks later, the mice were sacrificed, and the lungs and livers were collected for further analysis. All animal experiments were performed in accordance with the institutional ethical guidelines for animal care, and approved by the Animal Experimentation Ethics Committee of Nantong University (Approval ID: SYXK (SU) 2017-0046).

## Statistical analysis

Statistical significance of differences between groups were tested by using analysis of variance (ANOVA) or unpaired Student's t-test. Data were presented as mean ± SD from at least three independent experiments. All statistical analyses were performed using GraphPad Prism version 8.0 (GraphPad Software, La Jolla, CA, USA). *p<0.05 was accepted as statistical significance.

## Acknowledgements

We would like to thank Friedhelm Hildebrandt from University of Michigan Drive for providing the plasmid expressing PLCE1.

# Additional information

### Funding

| Funder | Grant reference number | Author |
|---|---|---|
| National Natural Science Foundation of China | 32271193 | Cheng Sun |
| Natural Science Foundation of Shanghai | 21ZR1449800 | Hui Shi |
| Scientific Project from Shanghai Municipal Health Commission | 202240015 | Hui Shi |
| Postgraduate Research & Practice Innovation Program of Jiangsu Province | KYCX23_3411 | Qianqian Ju |

The funders had no role in study design, data collection and interpretation, or the decision to submit the work for publication.

### Author contributions

Qianqian Ju, Formal analysis, Validation, Investigation, Methodology, Writing – original draft; Wenjing Sheng, Resources, Investigation; Meichen Zhang, Investigation, Methodology; Jing Chen, Resources, Validation; Liucheng Wu, Resources; Xiaoyu Liu, Methodology; Wentao Fang, Conceptualization,

Resources; Hui Shi, Conceptualization, Project administration; Cheng Sun, Conceptualization, Supervision, Funding acquisition, Writing – review and editing

**Author ORCIDs**
Qianqian Ju ⓘ https://orcid.org/0000-0001-7561-1058
Cheng Sun ⓘ https://orcid.org/0000-0001-8411-4619

**Ethics**
Human esophageal squamous tumor specimens were obtained from the Affiliated Hospital of Nantong University. All samples were collected with written informed consent from patients and in compliance with the ethical principles of the Declaration of Helsinki. The study protocol was approved by the Institutional Ethics Committee of the Affiliated Hospital of Nantong University (approval number: 2015 L132).

All animal experiments were performed in accordance with the institutional ethical guidelines for animal care, and approved by the Animal Experimentation Ethics Committee of Nantong University (Approval ID: SYXK (SU) 2017-0046).

Reviewer #1 (Public Review): https://doi.org/10.7554/eLife.97373.3.sa1
Reviewer #2 (Public Review): https://doi.org/10.7554/eLife.97373.3.sa2
Reviewer #3 (Public Review): https://doi.org/10.7554/eLife.97373.3.sa3
Author response https://doi.org/10.7554/eLife.97373.3.sa4

## Additional files

### Supplementary files
Supplementary file 1. he sequences used in gene knockdown and mutation.

Supplementary file 2. Primer sequences used in quantitative real time-PCR (qRT-PCR).

MDAR checklist

### Data availability
The dataset regarding MS-based protein phosphorylation modifications can be found in figshare https://doi.org/10.6084/m9.figshare.25271140.

The following dataset was generated:

| Author(s) | Year | Dataset title | Dataset URL | Database and Identifier |
|---|---|---|---|---|
| Ju Q | 2024 | LC-MS/MS spectrometry analysis of PLCE1 | https://doi.org/10.6084/m9.figshare.25271140 | figshare, 10.6084/m9.figshare.25271140 |

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

# Appendix 1

## Appendix 1—key resources table

| Reagent type (species) or resource | Designation | Source or reference | Identifiers | Additional information |
|---|---|---|---|---|
| Gene (*Homo sapiens*) | *Map3k7* | GenBank | Gene ID:6885 | |
| Gene (*Homo sapiens*) | *Plce1* | GenBank | Gene ID: 51196 | |
| Strain, strain background (*Escherichia coli*) | DH5α | Vazyme | C502 | Competent cells |
| Cell line (*Homo sapiens*) | ECA-109 | SIBCB | TCHu69 | |
| Cell line (*Homo sapiens*) | KYSE-150 | SIBCB | TCHu236 | |
| Cell line (*Homo sapiens*) | TE-1 | SIBCB | TCHu 89 | |
| Cell line (*Homo sapiens*) | HEK-293 | ATCC | CRL-1573 | |
| Transfected construct (human) | *Map3k7* shRNA/ shNC | Hanbio | This paper | Lentiviral construct to transfect and express the shRNA. |
| Transfected construct (human) | *Plce1* shRNA/shNC | OBiO | This paper | Lentiviral construct to transfect and express the shRNA. |
| Transfected construct (human) | siRNA to *Map3k7* | RiboBio | This paper | Transfected construct (human) |
| Transfected construct (human) | siRNA to *Plce1* | RiboBio | This paper | Transfected construct (human) |
| Antibody | Anti-TAK1 (Rabbit monoclonal) | Cell Signaling Technology | Cat#: 5206 | WB (1:1000) |
| Antibody | Anti-phospho- TAK1 (Ser412) (Rabbit monoclonal) | Cell Signaling Technology | Cat#: 9339 | WB (1:1000) |
| Antibody | Anti-PLCE1 (Rabbit polyclonal) | Invitrogen | Cat#: PA5-100856 | WB (1:1000) |
| Antibody | Anti-phospho-PLCE1 (Ser1060) (Rabbit polyclonal) | Absin | This paper | WB (1:1000) IHC (1:1000) |
| Antibody | Anti-PKCα (Rabbit monoclonal) | Cell Signaling Technology | Cat#: 2056 | WB (1:1000) |
| Antibody | Anti-phospho-PKC (pan) (gamma Thr514) (Rabbit monoclonal) | Cell Signaling Technology | Cat#: 38938 | WB (1:1000) |
| Antibody | Anti-GSK-3β (Rabbit monoclonal) | Cell Signaling Technology | Cat#: 9315 | WB (1:1000) |
| Antibody | Anti-phospho-GSK-3β (Ser9) (Rabbit monoclonal) | Cell Signaling Technology | Cat#: 5558 | WB (1:1000) |
| Antibody | Anti-β-Catenin (Rabbit monoclonal) | Cell Signaling Technology | Cat#: 8480- | WB (1:1000) |
| Antibody | Anti-phospho-β-Catenin (Ser33/37/Thr41) (Rabbit monoclonal) | Cell Signaling Technology | Cat#: 9561 | WB (1:1000) |
| Antibody | Anti-MMP-2 (Rabbit monoclonal) | Cell Signaling Technology | Cat#: 87809- | WB (1:1000) |
| Antibody | Anti-Actin (Mouse monoclonal) | Cell Signaling Technology | Cat#: 3700- | WB (1:1000) |
| Antibody | Myc-Tag (Sepharose Bead Conjugate) | Cell Signaling Technology | Cat#: 55464 | IP (1:20) |
| Antibody | Epithelial-Mesenchymal Transition (EMT) antibody sample kit | Cell Signaling Technology | Cat#: 9782 | WB (1:1000) |

*Appendix 1 Continued on next page*

*Appendix 1 Continued*

| Reagent type (species) or resource | Designation | Source or reference | Identifiers | Additional information |
|---|---|---|---|---|
| Antibody | Anti-IKK (Rabbit monoclonal) | Cell Signaling Technology | Cat#: 61294 | WB (1:1000) |
| Antibody | Anti-phosphor-IKK (Rabbit monoclonal) | Cell Signaling Technology | Cat#: 2078 | WB (1:1000) |
| Antibody | Anti-JNK (Rabbit monoclonal) | Cell Signaling Technology | Cat#: 9252 | WB (1:1000) |
| Antibody | Anti-phosphor-JNK (Rabbit monoclonal) | Cell Signaling Technology | Cat#: 4668 | WB (1:1000) |
| Antibody | Anti-ERK (Rabbit monoclonal) | Cell Signaling Technology | Cat#: 9102 | WB (1:1000) |
| Antibody | Anti-phospho-ERK (Rabbit monoclonal) | Cell Signaling Technology | Cat#: 4370 | WB (1:1000) |
| Antibody | Anti-P38 MPAK (Rabbit monoclonal) | Cell Signaling Technology | Cat#: 9212 | WB (1:1000) |
| Antibody | Anti-phospho-P38 MAPK (Rabbit monoclonal) | Cell Signaling Technology | Cat#: 9211 | WB (1:1000) |
| Antibody | Anti-normal rabbit IgG (Rabbit monoclonal) | Cell Signaling Technology | Cat#: 2729 | WB (1:1000) |
| Recombinant DNA reagent | pcDNA3.1-*Map3k7* (plasmid) | This paper | This paper | |
| Recombinant DNA reagent | pRK5-myc *Plce1* (plasmid) | This paper | Dr. Friedhelm Hildebrandt (1) | |
| Sequence-based reagent | *Map3k7* siRNA | RiboBio | This paper | GGAGTTGTTTGCAAAGCTA |
| Sequence-based reagent | *Plce1* siRNA-1 | RiboBio | This paper | GGACTTCAATATCGCAGTA |
| Sequence-based reagent | *Plce1* siRNA-2 | RiboBio | This paper | GTCGAAGTGTAGAATTGGA |
| Sequence-based reagent | *Plce1* siRNA-3 | RiboBio | This paper | CAATCATCATATCGATTGA |
| Sequence-based reagent | *Map3k7* gRNA_F | This paper | PCR primers | ccgAGGGGCTTCGATCATCTCAC |
| Sequence-based reagent | *Map3k7* gRNA_R | This paper | PCR primers | aacGTGAGATGATCGAAGCCCCT |
| Sequence-based reagent | *Plce1* (S1060A) _F | This paper | PCR primers | TGGAGTGCTCGAAACCCCG CACCCGGAACATCAGCAAA |
| Sequence-based reagent | *Plce1* (S1060A) _R | This paper | PCR primers | GGGGTTTCGAGCACTCCAC CGTCTGCCACCAAACAA |
| Sequence-based reagent | *Map3k7*_F | This paper | PCR primers | ATTGTAGAGCTTCGGCAGTTATC |
| Sequence-based reagent | *Map3k7*_R | This paper | PCR primers | CTGTAAACACCAACTCATTGCG |
| Sequence-based reagent | *Plce1*_F | This paper | PCR primers | GGGTGACATGGCTGATCCTC |
| Sequence-based reagent | *Plce1*_R | This paper | PCR primers | GACAGCGTTGTAGTTGCCCA |
| Sequence-based reagent | *Cdh1*_F | This paper | PCR primers | GCTTTACTGTTTCTCAAGTGT |
| Sequence-based reagent | *Cdh1*_R | This paper | PCR primers | AATACACAATTATCAGCACCC |

*Appendix 1 Continued on next page*

*Appendix 1 Continued*

| Reagent type (species) or resource | Designation | Source or reference | Identifiers | Additional information |
|---|---|---|---|---|
| Sequence-based reagent | Vim_F | This paper | PCR primers | AACTTCTCAGCATCACGAT |
| Sequence-based reagent | Vim_R | This paper | PCR primers | GTAGGAGTGTCGGTTGTT |
| Sequence-based reagent | Ctnnb1_F | This paper | PCR primers | AGAATTGAGTAATGGTGTAGAAC |
| Sequence-based reagent | Ctnnb1_R | This paper | PCR primers | TACCCATACATATCCCAAATAGT |
| Sequence-based reagent | Cldn1_F | This paper | PCR primers | TGTATAGTCCTCTTGGGTTG |
| Sequence-based reagent | Cldn1_R | This paper | PCR primers | AATTGTCAGTGGAGTCAGT |
| Sequence-based reagent | Cdh2_F | This paper | PCR primers | AAAAGGAAAGGAAAGAAAGGG |
| Sequence-based reagent | Cdh2_R | This paper | PCR primers | GTCAGAGGTGTATCATTTATATTCT |
| Sequence-based reagent | Zeb1_F | This paper | PCR primers | GTTGCTCCTTCTTCCTGA |
| Sequence-based reagent | Zeb1_R | This paper | PCR primers | ATGTGGTTCCTGTTCCTAG |
| Sequence-based reagent | Tjp1_F | This paper | PCR primers | TGTGGACATCCTACTTACTTAA |
| Sequence-based reagent | Tjp1_R | This paper | PCR primers | GAGAAGATAAAGAAACTGTTGTATG |
| Sequence-based reagent | Snai1_F | This paper | PCR primers | AGCTATTTCAGCCTCCTG |
| Sequence-based reagent | Snai1_R | This paper | PCR primers | TGTAAACATCTTCCTCCCAG |
| Sequence-based reagent | Snai2_F | This paper | PCR primers | CTGTATGAAACTGAGATGTTGT |
| Sequence-based reagent | Snai2_R | This paper | PCR primers | GAAGCAAGTAAAGTCTCTGAAA |
| Sequence-based reagent | Gapdh_F | This paper | PCR primers | AAGAGCACAAGAGGAAGAG |
| Sequence-based reagent | Gapdh_R | This paper | PCR primers | TAACTGGTTGAGCACAGG |
| Commercial assay or kit | PLC Activity Assay Kit | Jining Shiye | ml076627 | ELISA kit |
| Commercial assay or kit | Fluo-4 AM Assay Kit | Beyotime | S1060 | |
| Commercial assay or kit | Human Inositol 1,4,5-triphosphate enzyme-linked Immunosorbent Assay | Mlbio | ml060362 | ELISA kit |
| Commercial assay or kit | Human Diacylglycerol commercial ELISA Kit | Mlbio | ml026857 | ELISA kit |
| Chemical compound, drug | 5Z-7-oxozeaenol | Sigma-Aldrich | O9890 | TAK1 inhibitor |
| Chemical compound, drug | NG25 | Sigma-Aldrich | SML1332 | TAK1 inhibitor |
| Chemical compound, drug | Takinib | Sigma-Aldrich | SML2216 | TAK1 inhibitor |
| Chemical compound, drug | BAPTA-AM | MedChemExpress | HY-100545 | An intracellular calcium chelator |

*Appendix 1 Continued on next page*

*Appendix 1 Continued*

| Reagent type (species) or resource | Designation | Source or reference | Identifiers | Additional information |
|---|---|---|---|---|
| Chemical compound, drug | Midostaurin | MedChemExpress | HY-10230 | A PKC inhibitor |
| Chemical compound, drug | 2-APB | MedChemExpress | HY-W009724 | An IP3R antagonist |
| Peptide, recombinant protein | EGF | Gibco | PHG0311 | |
| Software, algorithm | GraphPad Prism version 8.0 | GraphPad | | |

