## [Editor Report · eLife assessment]

This work provides **solid** evidence that Transforming Growth Factor β Activated Kinase 1 (TAK1) regulates esophageal squamous cell carcinoma (ESCC) tumor proliferation and metastasis. The findings are **valuable** to the field of molecular tumor biology in general and to the understanding of ESCC tumor invasiveness and metastatic potential.

---

## [Referee Report · Reviewer #1 (Public Review)]

Summary:

In previously published work, the authors found that Transforming Growth Factor β Activated Kinase 1 (TAK1) may regulate esophageal squamous cell carcinoma (ESCC) tumor cell proliferation via the RAS/MEK/ERK axis. They explore the mechanisms for TAK1 as a possible tumor suppressor, demonstrating phospholipase C epsilon 1 as an effector of tumor cell migration, invasion and metastatic potential.

They explore the mechanisms for TAK1 as a possible tumor suppressor, demonstrating phospholipase C epsilon 1 as an effector of tumor cell migration, invasion and metastatic potential.

Strengths:

The authors show in vitro that TAK1 overexpression reduces tumor cell migration and invasion while TAK1 knockdown promotes a mesenchymal phenotype (epithelial-mesenchymal transition) and enhances migration and invasion. To explore possible mechanisms of action, the authors focused on phospholipase C epsilon 1 (PLCE1) as a potential effector, having identified this protein in co-immunoprecipitation experiments. Further, they demonstrate that TAK1-mediated phosphorylation of PLCE1 is inhibitory. Each of the observations is supported by different experimental strategies, e.g. use of different approaches for knockdown (pharmacologic, RNA inhibition, CRISPR/Cas). Xenograft experiments showed that suppression/loss of TAK1 is associated with more frequent metastases and conversely that PLCE1 is associated positively with xenograft metastases. A considerable amount of experimental data is presented for review, including supplemental data, that show that TAK1 regulation may be important in ESCC development.

Weaknesses:

As noted by the authors, immunoprecipitation (IP) experiments identified a number (24) of proteins as potential targets for the TAK1 ser/thr kinase. Prior work (cited as Shi et al, 2021) focused on a different phosphorylation target for TAK1, Ras association domain family 9 (RASSF9), but a more comprehensive discussion of the co-IP experiments would help place this work in better context.

---

## [Referee Report · Reviewer #2 (Public Review)]

Summary:

In this study, Ju Q et al performed both in vitro and in vivo experiments to test the effect of TAK1 on cancer metastasis. They demonstrated that TAK1 is capable of directly phosphorylating PLCE1 and this modification represses its enzyme activity, leading to suppression of PIP2 hydrolysis and subsequently signal transduction in the PKC/GSK-3β/β-Catenin axis.

Strengths:

The quality of data is good, and the presentation is well organized in a logical way.

Weaknesses:

The study missed some key link in connecting the effect of TAK1 on cancer metastasis via phosphorylating PLCE1.

---

## [Referee Report · Reviewer #3 (Public Review)]

Summary:

The research by Qianqian Ju et al. found that the knockdown of TAK1 promoted ESCC migration and invasion, whereas overexpression of TAK1 resulted in the opposite outcome. These in vitro findings could be recapitulated in a xenograft metastasis mouse model.

Mechanistically, TAK1 phosphorylates PLCE1 S1060 in the cells, decreasing PLCE1 enzyme activity and repressing PIP2 hydrolysis. As a result, reducing DAG and inositol IP3, thereby suppressing signal transduction of PKC/GSK 3β/β Catenin. Consequently, cancer metastasis-related genes were impeded by TAK1.

Overall, this study offers some intriguing observations. Providing a potential druggable target for developing agents for dealing with ESCC.

The strengths of this research are:

(1) The research uses different experimental approaches to address one question. The experiments are largely convincing and appear to be well executed.

(2) The phenotypes were observed from different angles: at the mouse model, cellular level, and molecular level.

(3) The molecular mechanism was down to a single amino acid modification on PLCE1.

The weaknesses part of this research are:

Most of the experiments were done in protein overexpression conditions, with the protein level increasing hundreds of folds in the cell, producing an artificial environment that would sometimes generate false positive results.

---

## [Author Response]

The following is the authors’ response to the original reviews.

**Recommendations for the authors:**

We would like to see the reviewers' critiques be addressed satisfactorily.

**Reviewer #1 (Recommendations For The Authors):**
While the manuscript reads fairly well, there are a number of minor grammatical edits that would improve the reading of this paper.

To improve the reading, we sent our manuscript out for language polishing using Wiley Editing Services. The changes were labeled in Red color.

The opening paragraph, while seeking to establish clinical relevance, likely can be removed or tailored.

We agreed with this concern, the first paragraph was tailored in the revised manuscript.

**Reviewer #2 (Recommendations For The Authors):**
Although the authors provided a substantial amount of data to support the conclusion, there are several important issues to be added to strengthen the study, as highlighted below:Figure 2: In this figure, the authors provided evidence that TAK1 phosphorylates PLCE1 at serine 1060. To make the data more convincing, the authors need to perform an in vitro kinase assay to confirm this result. Ideally, the in vitro kinase assay also includes a mutant form of PLCE1-S1060A as a control.

Thank the referee for this constructive comment. Since we cannot perform experiments with radioactive compounds in our institute, therefore the phosphorylation of PLCE1 at serine 1060 induced by TAK1 cannot be further confirmed by a routine in vitro kinase, in which 32P was used. Instead, we performed TAK1 and PLCE1 pulldown, and incubated these two proteins in a kinase assay buffer. The resulting samples were analyzed by western blot. Our data showed that TAK1 phosphorylates PLCE1 at serine 1060, as evidenced by a strong band for p-PLCE1 S1060 when TAK1 incubated with PLCE1. For the sample contained TAK1 and PLCE1 S1060A, the band density for p-PLCE1 S1060 was largely decreased. Ideally, there should be no band for p-PLCE1 S1060 when TAK1 incubated with PLCE1 S1060A. However, our current data detected p-PLCE1 S1060 in this reaction, although it was decreased as compared to wild type PLCE1. The reason for this is likely due to the presence of endogenous wild type PLCE1 in the TAK1 pull-down samples. These data were presented as Figure S6C in the revised manuscript.

Figure 4: In this part of the study, the author claimed that TAK1 inhibits PLCE1 enzyme activity. However, they fall short of evidence that this inhibitory effect of TAK1 on PLCE1 enzyme activity is mediated via phosphorylation at S1060.

Thank the referee for this critical comment. Actually, we measured the effect of TAK1 on mutate PLCE1 activity, which was presented in Figure 4B. The data showed that TAK1 has no inhibitory effect on PLCE1 S1060A enzyme activity. In contract, TAK1 repressed wild type PLCE1 activity (Figure 4A). These data indicate that, at least in part, the inhibitory effect of TAK1 on PLCE1 enzyme activity is mediated via phosphorylation at S1060.

Figures 6 and 7: Here the authors used ESCC metastasis model in nude mice to establish the role of TAK1 and PLCE1, respectively. However, the effects of TAK1 and PLCE1 are studied separately, and there no link to show that TAK1 inhibits metastasis via activation of PLCE1. Ideally the authors should use the transgenic mice with expression of mutant PLCE1-S1060A to support the conclusion.

We agreed with this notion that the transgenic mice with expression of mutant PLCE1-S1060A will further strengthen our conclusions. However, due to limited time and resource, we cannot generate such genetic mice. Thank the referee for this insightful and critical comment.

**Reviewer #3 (Recommendations For The Authors):**
(1) Have the authors ever checked the phosphorylation status of endogenous PLCE1 S1060p in the TAK1 overexpression alone ECA-109 cell line? Does it increase? Similarly, in siMap3k7 ECA-109 cells, does endogenous PLCE1 S1060p reduce?

Thank the referee for these critical comments. During the revision, we examined whether TAK1 overexpression or knockdown affects endogenous p-PLCE1 S1060 in ECA-109 cells. Our data showed that TAK1 overexpression induced an increase in p-PLCE1 S1060, whereas TAK1 knockdown resulted in a decrease in p-PLCE1 S1060. These data were presented in Figure S6A, B.

(2) The authors show that using TAK1 inhibitors cannot completely abolish all the phosphorylation of PLCE1 S1060 in cells and mice. Does it mean some other potential kinases also target PLCE1 S1060?

Thank the referee for this insightful comment. As mentioned by the referee, TAK1 inhibitors cannot completely abolish all the phosphorylation of PLCE1 S1060 in cells and mice. Therefore, it is likely that some other potential kinases also target PLCE1 S1060, we added this notion in the Discussion in the revised manuscript.

(3) PLCE1 S1060A completely bans the migration and invasion regulation function of TAK1 (Figure S10), indicating that PLCE1 S1060 is a very unique downstream target of TAK1 in migration and invasion regulation in the ECA-109 cell line. As a MAP3K, TAK1 was documented to regulate migration and invasion through multiple signal transduction pathways such as IKK, JNK, p38 MAPK, et al. Have the authors ever tried to test the effect of overexpression/knockdown of TAK1 on a few of these pathways in the ECA-109 cell line?

Thank the referee for these constructive comments. During the revision, we analyzed the effects of TAK1 on IKK, JNK, p38 MAPK, and ERK. Our data showed that TAK1 positively regulates these signal transduction pathways. For example, TAK1 overexpression increased p-IKK, p-JNK, p-P38 MAPK, and p-ERK in ECA-109 cells, whereas TAK1 knockdown decreased these protein levels. Although these pathways are affected by TAK1, with respect to cell migration and invasion, PLCE1 is likely a unique substrate of TAK1 in migration and invasion regulation in ECA-109 cells. We added these contents in the Results section in revised manuscript, and these data were presented in Figure S12A-D.

(4) Does TAK1 only catalyze the S1060 site on PLCE1 protein?

Thank the referee for this insightful comment. Currently, we just found TAK1 catalyze the S1060 site on PLCE1 protein, which cannot exclude the possibility that TAK1 also phosphorylates other residues on PLCE1 protein.

(5) Is there any PLCE1 S1060 point mutation existing in ESCC patients? Does it influence the prognosis of ESCC patients?

Thank the referee for this critical and constructive comment, which would further strengthen the significance of current study. However, we are facing a shortage of enough patient tumor samples for addressing this very important issue.

(6) What's the effect of TAK1 inhibitor on mice body weight?

Thank the referee for this critical comment. Since body weight is an important parameter, we measured mouse body weight during the whole experiments. The results showed that the body weight growth rate is not affected by TAK1 inhibitor, Takinib. These data were included in the revised manuscript as Figure S20A.

(7) For the control groups of the mouse xenograft tumor model in Figures 6 vs 7, why does the number of metastases behave so differently?

In Figure 6, the control mice were administered with ECA-109 cells via tail vein injection, mice were then treated with vehicle (saline). As for the control mice in Figure 7, they were administered with ECA-109 cells via tail vein injection. It should be mentioned that these cells were transduced with control lentivirus. Due to these differences, therefore, these two control mice have different number of metastases.